# Transferring Causal Effects using Proxies

**Manuel Iglesias-Alonso**[*]
ETH Zürich

**Felix Schur**[†]
Seminar for Statistics
ETH Zürich

**Julius von Kügelgen**[†]
Seminar for Statistics
ETH Zürich

**Jonas Peters**[†]
Seminar for Statistics
ETH Zürich

## Abstract

We consider the problem of estimating a causal effect in a multi-domain setting. The causal effect of interest is confounded by an unobserved confounder and can change between the different domains. We assume that we have access to a proxy of the hidden confounder and that all variables are discrete or categorical. We propose methodology to estimate the causal effect in the target domain, where we assume to observe only the proxy variable. Under these conditions, we prove identifiability (even when treatment and response variables are continuous). We introduce two estimation techniques, prove consistency, and derive confidence intervals. The theoretical results are supported by simulation studies and a real-world example studying the causal effect of website rankings on consumer choices.

## 1 Introduction

Estimating the causal effect of a treatment or exposure $X$ on an outcome or response $Y$ of interest is a core objective across the social and natural sciences [1–4]. However, a major obstacle to such causal inference from observational (passively collected) data is the existence of unmeasured confounders $U$, i.e., unobserved variables that influence both the treatment and the outcome. For example, suppose that we wish to study the causal effect of a new experimental drug $X$ on survival $Y$. Since such drugs tend to be administered only in severe cases, i.e., to patients in poor health with an already dire prognosis $U$, untreated patients will generally have better health outcomes. Naïvely comparing average survival between treated and untreated patients can thus lead to erroneous conclusions [5]. For accurate causal inference, it is therefore important that confounders are measured and correctly adjusted for [6].

Since it is difficult to rule out that some confounders remain unobserved, randomised controlled trials (RCTs) are considered the gold standard for causal inference [7, 8]. In an RCT, the treatment assignment is randomised, thereby eliminating any influences from potential confounders. However, for most settings of interest, RCTs are infeasible due to ethical or practical constraints. Much research has thus been dedicated to devising specialised methods for causal inference from purely observational data. An important step in this endeavour is to establish identifiability, i.e., to show that the causal estimand can be expressed in terms of distributions that only involve observed quantities [9]. To this end, existing methods leverage additional structure and assumptions, often in the form of other observed variables, such as instruments [10], mediators [11], or proxies [12, 13] (sometimes called negative controls [14]), which have certain causal relationships to the treatment, outcome, and confounder. In the present work, we propose a novel proxy-based approach.

---

[*]Not with ETH anymore, but most of this work was done while MIA was a student at this university. Correspondence to: igalonsomanuel@gmail.com.

[†]Joint supervision.

39th Conference on Neural Information Processing Systems (NeurIPS 2025).

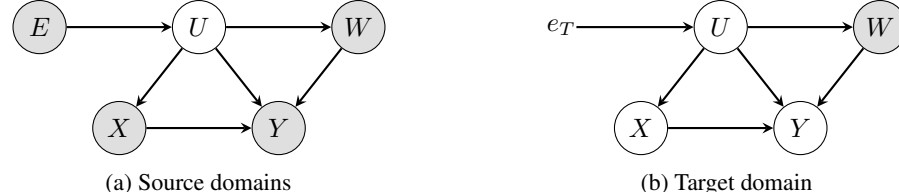

(a) Source domains                 (b) Target domain

Figure 1: **Causal effect estimation in unseen domains using proxies.** We seek to estimate the causal effect of a treatment $X$ on an outcome $Y$ in the presence of an unobserved confounder $U$. The main learning signal takes the form of proxy measurements $W$ of $U$. Moreover, we observe data from multiple domains or environments $E$, which differ through shifts in the distribution of $U$. (a) In the available source domains, we observe $E$, $X$, $Y$, and $W$. (b) In the target domain ($E = e_T$) for which we aim to estimate the causal effect, only $W$ is observed. We prove that the available data can suffice to identify the target interventional distribution $\mathbb{Q}_Y^{\mathrm{do}(X:=x)} := \mathbb{P}_Y^{\mathrm{do}(X:=x, E:=e_T)}$.

## 1.1 Related work

The intuition behind proximal causal inference is that, even if some confounders $U$ are unobserved and thus cannot be directly adjusted for, observed proxy variables $W$ that are related to $U$ may provide sufficient information to correct for the unobserved confounding [15]. In contrast to approaches rooted in latent variable modelling [e.g., 16–18], proximal causal inference methods typically do not aim to estimate the confounder explicitly but instead seek to devise more direct estimation methods. In contrast to literature on transportability and data fusion [e.g. 19–21] where identifiability is determined based purely on graphical criteria, proxy-based methods often rely on additional assumptions concerning the informativeness of the available proxies.

**Single noisy measurement of the confounder.** In the simplest case, we observe a single proxy $W$ of $U$ which is independent of $X$ and $Y$ given $U$ (e.g., resulting from non-differential measurement error; see inset figure), and both $U$ and $W$ are discrete [22–24]. In this case, the confounding bias can be reduced (compared to no adjustment) by adjusting for $W$ if the confounder acts monotonically [25]. Further, if the error mechanism, 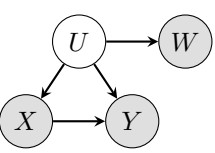 i.e., the conditional distribution $\mathbb{P}(W|U)$ is known, the causal effect is fully identified via the matrix adjustment method of Greenland and Lash [26]. However, in practice, $\mathbb{P}(W|U)$ is typically unknown, and proxies may causally influence the treatment or the outcome.

**Learning from multiple proxies.** As shown by Kuroki and Pearl [12], $\mathbb{P}(W|U)$ and thus the causal effect of $X$ on $Y$ can sometimes be identified if a second discrete proxy $Z$ of $U$ is available (see inset figure without the dashed edge), provided that the proxies are sufficiently informative (formalised via invertibility of certain matrices of conditional probabilities). 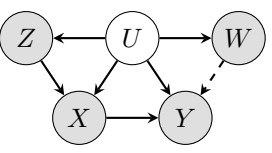 Miao et al. [27] extend this result to the more general setting in which $W$ can influence $Y$ (where $\mathbb{P}(W|U)$ is no longer identifiable) and establish conditions for identifiability when the confounder and proxies are continuous. For the latter, the matrix adjustment method is replaced by an approach based on bridge functions, obtained as the solutions to certain integral equations [27]. Subsequent work provides extensions to longitudinal settings [28, 29] and efficient estimation methods based on semiparametric models [30], kernel methods [31], or deep learning [32].

**Proxies for domain adaptation.** Alabdulmohsin et al. [33] use proxy methods for single-source unsupervised domain adaptation, i.e., to predict $Y$ in a new target domain where only $X$ is observed. To this end, they rely on the latent shift assumption, which posits that the domain shift is entirely due to a shift in the distribution of a discrete latent confounder $U$ of $X$ and $Y$, and leverage source observations of a noisy measurement $W$ of $U$ and a concept bottleneck variable [34] that mediates the effect of $X$ on $Y$ to identify the optimal target predictor. Tsai et al. [35] generalise these results in several ways, most notably showing that the concept bottleneck is not needed if multiple, sufficiently diverse source domains are available (see Fig. 1a) and both $W$ and $X$ are observed in the target domain. Prashant et al. [18] consider domain generalisation under the latent shift assumption via approximate identification of the unobserved confounder in the presence of either proxies or multiple domains.

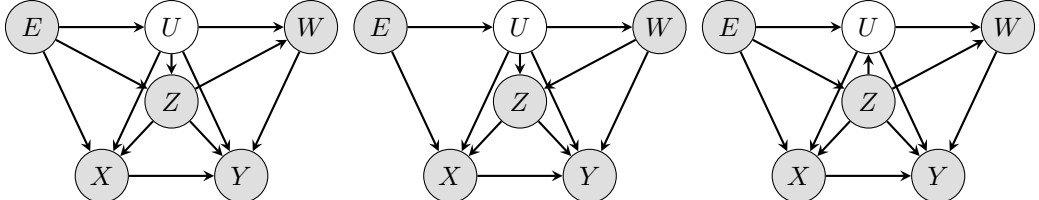

Figure 2: **Examples of causal graphs satisfying our identifiability conditions.** Whereas the paper mostly focuses on the scenario from Fig. 1, our identifiability results hold for a broader class of causal models including additional covariates $Z$ and unmediated covariate shift via $E \rightarrow X$, see Appx. B.

**Learning from multiple domains.** Causal approaches to domain generalization [e.g., 36–40] typically seek to discard spurious features and learn one or more domain-invariant predictors for $Y$, rather than the target interventional distribution $\mathbb{Q}(Y|\text{do}(x))$. In contrast to work on combining experimental and observational data to improve statistical efficiency [e.g., 41, 42], we assume no access to observations of the outcome $Y$ in the target domain.

## 1.2 Overview and main contributions

In this work, we consider the problem of estimating the causal effect of $X$ on $Y$ in the presence of a discrete unobserved confounder $U$, given a proxy $W$ of $U$ and data from multiple domains. Our main focus is on the setting illustrated in Fig. 1, which also relies on the latent shift assumption [33] (though see Fig. 2, § 3.1, and Remark 9 in Appx. B for a discussion of the relaxation of this assumption) and is similar to that considered by Tsai et al. [35]. However, a key difference is the estimand of interest. Whereas Tsai et al. seek to estimate the conditional mean $\mathbb{E}[Y|X = x]$ under the target distribution $\mathbb{Q}$, our focus is on the interventional distribution $\mathbb{Q}(Y|\text{do}(x))$. Moreover, we only need to observe $W$ in the target domain, as shown in Fig. 1b; we thus need to transfer information from the source domains to the target domain. While our goal is more aligned with proximal causal inference than with domain adaptation, a key difference to the setting studied, e.g., by Miao et al. [13] is that in our case, the causal effect of $X$ on $Y$ may differ across domains due to shifts in the distribution of $U$ (even though the joint causal effect of $X$ *and* $U$ on $Y$ is invariant [e.g., 36]). For ease of presentation, we focus on a setting in which all variables are discrete. We discuss some extensions to continuous variables in § 3.1 and § 7.

The remainder of the paper is organised as follows. First, we formally state the considered problem setting (§ 2) and establish sufficient conditions for identifiability (§ 3). We then propose two estimators based on alternative parametrisations of the model and study their statistical properties (§ 4). Empirically, we investigate our estimators in simulations (§ 5) and present an application to hotel ranking data (§ 6). We conclude with a summary and an outlook for future work (§ 7).

We highlight the following main contributions:

- We prove that, given only observations of $W$ in the target domain, the interventional distribution $\mathbb{Q}(Y|\text{do}(x))$ is identified (Thm. 1) if the available source domains are sufficiently diverse (Asm. 1). Furthermore, we relax the latent shift assumption by allowing for additional covariates and edges, for example (Fig. 2 and Thm. 8).

- We propose two consistent estimators, one of which is asymptotically normal and comes with confidence intervals (Props. 4 and 5).

- We empirically verify that our estimators compare favourably to naive and proxy-adjustment baselines (Fig. 4) and that the derived confidence intervals have valid coverage and decreasing length as sample size increases.

## 2 Intervention distributions under domain shifts using proxies

**Data generating process.** Formally, we assume that the data are generated by a structural causal model (SCM) [43, 44] $\mathcal{C} := (S, \mathbb{P}_N)$ with the following collection $S$ of assignments:

$$E := f_E(N_E), \qquad U := f_U(E, N_U), \qquad W := f_W(U, N_W), \tag{1}$$
$$X := f_X(U, N_X), \qquad Y := f_Y(U, W, X, N_Y).$$

The noise variables $N := (N_E, N_U, N_W, N_X, N_Y)$ follow the joint distribution $\mathbb{P}_N$ and are jointly independent. The SCM $\mathcal{C}$ induces the causal graph $\mathcal{G}$ shown in Fig. 1 and a distribution over the random variables $(E, U, W, X, Y)$, defined as the push-forward of $\mathbb{P}_N$ via (1), which we denote by $\mathbb{P}^\mathcal{C}$ or $\mathbb{P}$. As proved, e.g., by Lauritzen et al. [45], this induced distribution is Markov with respect to (w.r.t.) $\mathcal{G}$. (Since our work does not consider counterfactual statements, one could also use causal graphical models and assume that the observed distribution is Markov w.r.t. the graph in Fig. 1(a).)

**Generalization.** For sake of accessibility, we focus our presentation on the setting from Fig. 1. However, our identifiability results directly apply to a broader class of causal models (that may contain additional covariates $Z$), whose induced causal graphs $\mathcal{G}$ satisfy the d-separation [9] statements

$$Y \perp\!\!\!\perp_\mathcal{G} E | (U, X, Z), \quad W \perp\!\!\!\perp_\mathcal{G} (E, X) | (U, Z), \quad U \perp\!\!\!\perp_{\mathcal{G}_{\overline{X}}} X | Z, \quad Y \perp\!\!\!\perp_{\mathcal{G}_{\underline{X}}} X | (U, Z),$$

where $\mathcal{G}_{\overline{X}}$ and $\mathcal{G}_{\underline{X}}$ denote the graphs obtained by respectively removing the edges coming into $X$ and going out of $X$ from $\mathcal{G}$, see Fig. 2 for examples and Appx. B.4 for a more detailed discussion. We believe that our estimators and inference results can also be generalised to these settings analogously.

**Domains.** We consider $k_E$ source domains. The variable $E$ is a categorical domain indicator taking values in $\{e_1, \ldots, e_{k_E}\}$. The labels $e_i$ are arbitrary and serve only to distinguish domains. We assume no structure or prior knowledge about the domains. The target domain corresponds to $E = e_T$ (not necessarily contained in $\{e_1, \ldots, e_{k_E}\}$) and is described by the distribution $\mathbb{P}^{\mathcal{C};\mathrm{do}(E := e_T)}$, which we denote by $\mathbb{Q}^\mathcal{C}$ or $\mathbb{Q}$.

**Support.** Throughout, we assume that $E, U, W, X$, and $Y$ are discrete (though see the discussion after Thm. 1). We denote the support of a discrete random variable $V$ by $\mathrm{supp}(V) = \{v_1, \ldots, v_{k_V}\}$, so, e.g., $\mathrm{supp}(X) = \{x_1, \ldots, x_{k_X}\}$. We assume that $\mathbb{P}^\mathcal{C}$ has full support over $(E, U, W, X, Y)$, i.e., $\mathrm{supp}((E, U, W, X, Y)) = \mathrm{supp}(E) \times \mathrm{supp}(U) \times \mathrm{supp}(W) \times \mathrm{supp}(X) \times \mathrm{supp}(Y)$. The probability mass functions (pmfs) in the source and target domains are denoted by lowercase letters $p$ and $q$, respectively, so, e.g., $p(y|e, x) = \mathbb{P}(Y = y | E = e, X = x)$ or $q(w) = \mathbb{Q}(W = w)$.

**Matrix notation.** We write marginal pmfs as column vectors, $P(A) := (p(a_1), \ldots, p(a_{k_A}))^\top$, and conditional pmfs as matrices, $P(A|B) = (p(a_i|b_j))_{ij}$, with the $i^{\text{th}}$ row given by $P(a_i|B)$ and the $j^{\text{th}}$ column by $P(A|b_j)$. For additional discrete random variables $C$ and $D$, we write $P(a|B, c) := (p(a|b_1, c), \ldots, p(a|b_{k_B}, c))$ and $P(a|B, C, d) := (p(a|B, c_1, d)^\top, \ldots, p(a|B, c_{k_C}, d)^\top)$. In the target domain, we use $Q$ instead of $P$. The distributions induced by the last four structural assignments of the SCM $\mathcal{C}$ in Eq. (1) are thus described by the matrices $\{P(U|E), Q(U)\}$, $P(W|U)$, $P(X|U)$, and $\{P(y|U, W, x)\}_{x \in \mathrm{supp}(X), y \in \mathrm{supp}(Y)}$, respectively.

**Data.** As illustrated in Fig. 1, we assume that we obtain a sample of $n_{\mathrm{src}}$ independent and identically distributed (i.i.d.) source realizations $(E_1, W_1, X_1, Y_1), \ldots, (E_{n_{\mathrm{src}}}, W_{n_{\mathrm{src}}}, X_{n_{\mathrm{src}}}, Y_{n_{\mathrm{src}}})$ from $\mathbb{P}_O := \mathbb{P}^\mathcal{C}_{(E, W, X, Y)}$ and $n_{\mathrm{tgt}} := n - n_{\mathrm{src}}$ i.i.d. target realizations $W_{n_{\mathrm{src}}+1}, \ldots, W_n$ from $\mathbb{Q}$.

**Estimand.** The goal is to estimate

$$q(y|\mathrm{do}(x)) := \mathbb{Q}(Y = y | \mathrm{do}(X := x)), \tag{2}$$

which, by slight abuse of notation, we call the causal effect from $X$ on $Y$ (in the target domain $e_T$). Since the domain is allowed to affect the hidden confounder, the causal effect in the target domain is, in general, different from $p(y|e, \mathrm{do}(x))$ when $e \in \{1, \ldots, e_K\}$ is a source domain.

## 3  Identifiability

We now show that (2) is identifiable from $\mathbb{P}_O$ and $\mathbb{Q}_W$. More formally, under suitable assumptions, we prove the following statement: if $\mathcal{C}_1$ and $\mathcal{C}_2$ are of the form of (1) such that $\mathbb{P}^{\mathcal{C}_1}_O = \mathbb{P}^{\mathcal{C}_2}_O$ and $\mathbb{Q}^{\mathcal{C}_1}_W = \mathbb{Q}^{\mathcal{C}_2}_W$, then $q^{\mathcal{C}_1}(y|\mathrm{do}(x)) = q^{\mathcal{C}_2}(y|\mathrm{do}(x))$. The key idea of the proof (inspired by ideas presented by Miao et al. [27, Section 2]) is to consider the covariate adjustment formula [6, 46, 47]

$$q(y|\mathrm{do}(x)) = Q(y|U, x) \, Q(U) = \sum_{r=1}^{k_U} q(y|u_r, x) \, q(u_r). \tag{3}$$

Clearly, $U$ is unobserved, so we cannot exploit this equation directly. Instead, we will rewrite $Q(y|U, x)$, so that it depends on $U$ only through a factor $P(W|U)$. Since, by invariance or modularity, $P(W|U) = Q(W|U)$, the dependence on $U$ vanishes after marginalising w.r.t. $U$ as in Eq. (3). To do so, we use invertibility of $P(W|U)$, which is guaranteed by the following assumption:

**Assumption 1.** *For all $x \in \text{supp}(X)$, we have* $\text{rank}(P(W|E, x)) \geq k_U$.

Intuitively, Asm. 1 states that the proxy $W$ is sufficiently informative about the confounder $U$ in the sense that the domain shifts in $U$ induce sufficient variability in the conditional distribution of $W|X = x$. As we show in Prop. 6 in Appx. B.1, Asm. 1 implies that the proxy $W$ can be transformed in such a way that the matrix $P(W|E, x)$ has linearly independent rows. We therefore assume this without loss of generality (see Appx. B.1 for a more detailed treatment). Finally, this implies that the matrix $P(W|E, x) \, P(W|E, x)^\top$ is invertible [48], so its right pseudoinverse $P(W|E, x)^\dagger$ exists.[3] Similar assumptions about the rank of conditional probability matrices are common in proxy-based identifiability results [e.g. 12, 13, 35].

We are now able to state the main identifiability result. Its proof, together with other proofs of this work, can be found in Appx. A.

**Theorem 1.** *Under the data generating process described in § 2 (ignoring the paragraph "Generalization") and assuming Asm. 1, we have for all $x \in \text{supp}(X)$ and $y \in \text{supp}(Y)$:*

$$q(y|\text{do}(x)) = P(y|E, x) \, P(W|E, x)^\dagger \, Q(W). \tag{4}$$

*Therefore, if $(E, W, X, Y)$ is observed in the source domains and $W$ in the target domain, the causal effect of $X$ on $Y$ in the target domain $e_T$ is identifiable.*

**Remark 2.** *If, within our setting, Asm. 1 does not hold, identifiability is, in general, lost (see Appx. B.2 for a counterexample). Thus, in this sense, Asm. 1 is necessary for Eq. (4) to hold.*

### 3.1 Extensions of the identifiability result

**Relaxing assumptions of Thm. 1 and special cases.** The result still holds (with an analogous proof) if $X$ and $Y$ are not discrete but continuous. In this case, $P(y|E, x)$ is a vector of evaluations of a conditional probability density function (pdf), for example. Furthermore, we do not assume faithfulness, and Thm. 1 still holds if there is no edge between $W$ and $Y$. Moreover, the latent shift assumption [33] can be relaxed to also allow for (unmediated) covariate shift via $E \rightarrow X$, see Remark 9 in Appx. B.4 for a characterisation of the necessary graphical assumptions.

**Incorporating covariates/observed confounders.** In practice, we often observe additional covariates that we want to include in the model. For the case of observed confounders $Z$ of $X$ and $Y$, we show that the (target-domain) conditional causal effect $q(y|\text{do}(x), z)$ given $Z = z$ is identified if $Z$ is observed in both the source and target domains and subject to an assumption similar to Asm. 1 for the matrix $P(W|E, x, z)$, see Asm. 2 and Thm. 8 in Appx. B.4 for details.

**(Conditional) average treatment effects.** Thm. 1 directly implies that the (target-domain) average treatment effect (ATE), given by $\mathbb{E}[Y|\text{do}(X := 1)] - \mathbb{E}[Y|\text{do}(X := 0)]$ under $\mathbb{Q}$, is identifiable, too. Moreover, due to the identifiability of the causal effect conditional on observed confounders $Z = z$, we can also identify the (target-domain) conditional average treatment effect (CATE), given by $\mathbb{E}[Y|\text{do}(X := 1), Z = z] - \mathbb{E}[Y|\text{do}(X := 0), Z = z]$ under $\mathbb{Q}$.

**Continuous proxy.** In Appx. B.5, we discuss an extension to the case where the proxy $W$ is continuous. The proposed approach is based on the idea that suitable discretisation can yield a discrete proxy variable that satisfies Asm. 1, see Asm. 3 and Prop. 10 for details.

## 4 Estimation and inference

Having established the identifiability of $q(y|\text{do}(x))$, we now construct estimators for this causal effect. Specifically, we propose two estimators based on different decompositions of the causal effect,

---

[3]Given a matrix $A \in \mathbb{R}^{m \times n}$ s.t. $AA^\top$ is invertible, its right pseudoinverse is defined as $A^\dagger := A^\top (AA^\top)^{-1}$.

denoted by $\widehat{q}_{C,n}(y|\mathrm{do}(x))$ and $\widehat{q}_{R,n}(y|\mathrm{do}(x))$, respectively. The subscript $n$ indicates the sample size used in each case. By a slight abuse of notation, we use the same symbols to denote the estimators and their values evaluated on a specific dataset. Code implementing both estimators, along with the experiments, is available on https://github.com/manueligal/proxy-intervention.

## 4.1 Causal parametrisation

As discussed in § 2, the structural assignments in Eq. (1) can be represented by the matrices $P(U|E)$, $Q(U)$, $P(W|U)$, $P(X|U)$ and $\{P(y|U,W,x)\}_{x\in\mathrm{supp}(X),y\in\mathrm{supp}(Y)}$. For our first approach we explicitly parametrise these matrices, thereby capturing the underlying causal mechanisms. Prop. 3 shows that the estimand $q(y|\mathrm{do}(x))$ can be expressed in terms of a subset of these (unknown) matrices.

**Proposition 3.** *For all $x \in \mathrm{supp}(X)$ and $y \in \mathrm{supp}(Y)$ we have*[4]

$$q(y|\mathrm{do}(x)) = \mathrm{diag}\Big(P(y|U,W,x)\,P(W|U)\Big)\,Q(U). \tag{5}$$

We now proceed by maximum likelihood estimation. Consider a parameter $\theta$ whose components are the entries of $P(U|E)$, $Q(U)$, $P(W|U)$, $P(X|U)$ and $\{P(y|U,W,x)\}_{x\in\mathrm{supp}(X),y\in\mathrm{supp}(Y)}$. By an abuse of notation, we denote the entries of $\theta$ by the corresponding probabilities with a subscript $\theta$, i.e.,

$$\theta := \Big(p_\theta(u_1|e_1),\ldots,p_\theta(u_{k_U}|e_{k_E}),q_\theta(u_1),\ldots,q_\theta(u_{k_U}),p_\theta(w_1|u_1),\ldots,p_\theta(w_{k_W}|u_{k_U}),$$
$$p_\theta(x_1|u_1),\ldots,p_\theta(x_{k_X}|u_{k_U}),p_\theta(y_1|u_1,w_1,x_1),\ldots,p_\theta(y_{k_Y}|u_{k_U},w_{k_W},x_{k_X})\Big). \tag{6}$$

The entries are restricted to be non-zero, each column of $P(U|E)$, $Q(U)$, $P(W|U)$, and $P(X|U)$ sums to 1, and similarly $\sum_{i=1}^{k_Y} p(y_i|u,w,x) = 1$ for all $u \in \mathrm{supp}(U)$, $w \in \mathrm{supp}(W)$, and $x \in \mathrm{supp}(X)$. To ensure these constraints are satisfied without explicitly enforcing them during optimisation, we use real-valued parameters (representing logits) and transform them into valid pmfs using the softmax function.

Naturally, the joint distributions of the observed variables $\mathbb{P}_{W,X,Y|E}$ in the source domains and $\mathbb{Q}_W$ in the target domain are also parametrised by $\theta$. Specifically, for all $i \in \{1,\ldots,k_Y\}$, $j \in \{1,\ldots,k_W\}$, $s \in \{1,\ldots,k_X\}$, and $l \in \{1,\ldots,k_E\}$, we have that

$$p_\theta(y_i,x_s,w_j|e_l) := \sum_{r=1}^{k_U} p_\theta(y_i|u_r,w_j,x_s)\,p_\theta(w_j|u_r)\,p_\theta(x_s|u_r)\,p_\theta(u_r|e_l)$$

$$q_\theta(w_j) := \sum_{r=1}^{k_U} p_\theta(w_j|u_r)\,q_\theta(u_r),$$

where both of these equations hold for the true (conditional) pmfs. The (conditional) likelihood over the observed variables is thus given by

$$L(\theta) := \prod_{i=1}^{k_Y}\prod_{s=1}^{k_X}\prod_{j=1}^{k_W}\prod_{l=1}^{k_E} p_\theta(y_i,x_s,w_j|e_l)^{n(y_i,x_s,w_j,e_l)} \prod_{j=1}^{k_W} q_\theta(w_j)^{n(w_j)}, \tag{7}$$

where $n(y_i,x_s,w_j,e_l)$ denotes the number of observations for the tuple $(y_i,x_s,w_j,e_l)$ in the source domains and $n(w_j)$ denotes the number of observations of $w_j$ in the target domain.

For all $x \in \mathrm{supp}(X)$ and $y \in \mathrm{supp}(Y)$, we define the function

$$g_{x,y} : \theta \longmapsto g_{x,y}(\theta) := \mathrm{diag}\Big(P_\theta(y|U,W,x)\,P_\theta(W|U)\Big)\,Q_\theta(U), \tag{8}$$

where the entries of these matrices are components of $\theta$. We denote by $\theta_0$ the true value of $\theta$, so by Prop. 3 we have that $g_{x,y}(\theta_0) = q(y|\mathrm{do}(x))$. Therefore, if we denote by $\widehat{\theta}_n$ a maximizer of

---

[4]The diag operator applied to a square matrix $A = (A_{i,j})_{1\le i,j\le n} \in \mathbb{R}^{n\times n}$ is defined as $\mathrm{diag}(A) := (A_{1,1}, A_{2,2},\ldots,A_{n,n}) \in \mathbb{R}^n$.

the likelihood function in Eq. (7), we define the plug-in estimator of the causal effect in the causal parametrisation for all $x \in \mathrm{supp}(X)$ and $y \in \mathrm{supp}(Y)$ as $\widehat{q}_{C,n}(y|\mathrm{do}(x)) \coloneqq g_{x,y}(\widehat{\theta}_n)$.

Even if the estimator $\widehat{\theta}_n$ does not converge, its induced probabilities over the observed variables concentrate around the true values, and the estimator $\widehat{q}_{C,n}(y|\mathrm{do}(x))$ is consistent.

**Proposition 4.** *Suppose that Asm. 1 holds. The estimator $\widehat{q}_{C,n}(y|\mathrm{do}(x))$ is consistent, that is,*

$$\widehat{q}_{C,n}(y|\mathrm{do}(x)) \;\to\; q(y|\mathrm{do}(x)) \text{ in probability as } n_{\mathrm{src}}, n_{\mathrm{tgt}} \to \infty.$$

### 4.2 Reduced parametrisation

Although $q(y|\mathrm{do}(x))$ is identifiable, not all the components of the vector $\theta$ that we optimise in the causal parametrisation can be identified from the observed data – the estimator is based on an overparametrisation of the problem. Our second approach estimates only the components necessary to compute Eq. (4). To do so, for all $x \in \mathrm{supp}(X)$ and $y \in \mathrm{supp}(Y)$, we consider one long parameter vector $\eta_{x,y}$ describing all entries of[5] $Q(W, e_T)$, $q(e_T)$, $P(W, x, E)$, $P(y, x, E)$, and $P(x, E)$ (see Appx. B.3 for details). Given an $\eta_{x,y}$, we can then not only form the matrices $Q_{\eta_{x,y}}(W, e_T)$, $q_{\eta_{x,y}}(e_T)$, $P_{\eta_{x,y}}(W, x, E)$, $P_{\eta_{x,y}}(y, x, E)$ and $P_{\eta_{x,y}}(x, E)$ but also $Q_{\eta_{x,y}}(W)$, $P_{\eta_{x,y}}(W|E, x)$, and $P_{\eta_{x,y}}(y|E, x)$, which we use to define an estimator based on Eq. (4). The components of $\eta_{x,y}$ are probabilities, so we can estimate their true value by empirical frequencies[6], that is, $\widehat{\eta}_{x,y,n} \coloneqq \frac{1}{n} \sum_{i=1}^{n} \eta_{x,y}^i$, where, for all $i \in \{1, \dots, n\}$,

$$\begin{aligned}
\eta_{x,y}^i \coloneqq \Big( & \mathbb{1}(W_i = w_1, E_i = e_T), \dots, \mathbb{1}(W_i = w_{k_W - 1}, E_i = e_T), \mathbb{1}(E_i = e_T), \\
& \mathbb{1}(W_i = w_1, X_i = x, E_i = e_1), \dots, \mathbb{1}(W_i = w_{k_W - 1}, X_i = x, E_i = e_{k_E}), \\
& \mathbb{1}(Y_i = y, X_i = x, E_i = e_1), \dots, \mathbb{1}(Y_i = y, X_i = x, E_i = e_{k_E}), \\
& \mathbb{1}(X_i = x, E_i = e_1), \dots, \mathbb{1}(X_i = x, E_i = e_{k_E}) \Big).
\end{aligned} \tag{9}$$

Then, for example,

$$\big(P_{\widehat{\eta}_{x,y,n}}(W|E, x)\big)_{j,l} = \frac{\frac{1}{n} \sum_{i=1}^{n} \mathbb{1}(W_i = w_j, X_i = x, E_i = e_l)}{\frac{1}{n} \sum_{i=1}^{n} \mathbb{1}(X_i = x, E_i = e_l)}. \tag{10}$$

For all $x \in \mathrm{supp}(X)$ and $y \in \mathrm{supp}(Y)$, we define the function

$$h : \eta_{x,y} \longmapsto h(\eta_{x,y}) \coloneqq P_{\eta_{x,y}}(y|E, x) \, P_{\eta_{x,y}}(W|E, x)^\dagger \, Q_{\eta_{x,y}}(W), \tag{11}$$

and define the estimator of the causal effect in the reduced parametrisation as

$$\widehat{q}_{R,n}(y|\mathrm{do}(x)) \coloneqq h(\widehat{\eta}_{x,y,n}). \tag{12}$$

The following proposition proves consistency of $\widehat{q}_{R,n}(y|\mathrm{do}(x))$ and provides confidence intervals.

**Proposition 5.** *Suppose that Asm. 1 holds. For all $x \in \mathrm{supp}(X)$ and $y \in \mathrm{supp}(Y)$, the estimator $\widehat{q}_{R,n}(y|\mathrm{do}(x))$ defined in Eq. (12) is a consistent estimator of $q(y|\mathrm{do}(x))$. Furthermore,*

$$\frac{\sqrt{n}}{\widehat{\sigma}_{x,y}} \big( \widehat{q}_{R,n}(y|\mathrm{do}(x)) - q(y|\mathrm{do}(x)) \big) \xrightarrow{\mathcal{D}} \mathcal{N}(0, 1),$$

*where*

$$\widehat{\sigma}_{x,y}^2 \coloneqq \nabla h(\widehat{\eta}_{x,y,n})^\top \, \widehat{\Sigma}_{x,y} \, \nabla h(\widehat{\eta}_{x,y,n}),$$

*with $\widehat{\Sigma}_{x,y}$ being the sample covariance matrix of $(\eta_{x,y}^1, \dots, \eta_{x,y}^n)$.*

---

[5]To simplify notation (for inference), in this section we model the sample as an i.i.d. sample $(E_1, W_1, X_1, Y_1), \dots, (E_n, W_n, X_n, Y_n)$ with $E_i \in \{e_1, \dots, e_{k_E}\} \cup \{e_T\}$; whenever $E_i = e_T$, the entries of $X$ and $Y$ are missing. This way, $n$ is fixed, while $n_{\mathrm{src}} \coloneqq \sum_i \mathbb{1}(E_i \neq e_T)$ and $n_{\mathrm{tgt}} \coloneqq \sum_i \mathbb{1}(E_i = e_T)$ are random. For all $j \in \{1, \dots, k_W\}$, $q(w_j) = q(w_j|e_T)$ can now also be considered a conditional probability.

[6]Strictly speaking, we consider a slightly modified version: on the events where the matrix in Eq. (10) does not have linearly independent rows, we change $\widehat{\eta}_{x,y,n}$ to ensure that it does. This happens with probability vanishing to zero (see proof of Prop. 5) but ensures that Eq. (12) is always well-defined.

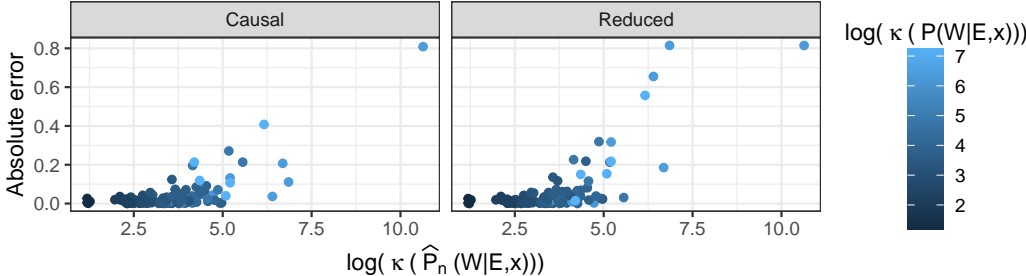

Figure 3: **Absolute estimation error increases for near non-invertible** $P(W|E, x)$. For $M{=}25$ distinct data generating processes, we draw $N{=}5$ datasets each, consisting of $k_E{=}2$ source domains and $n{=}20\,000$ realizations. The errors for both estimators increase with the condition number $\kappa$ of the matrix $P(W|E, x)$ (or its estimated counterpart), a measure of its non-invertibility.

By Prop. 5, pointwise asymptotic confidence intervals for $q(y|\mathrm{do}(x))$ at level $(1 - \alpha)$ are given by

$$\left[ \widehat{q}_{R,n}(y|\mathrm{do}(x)) - \frac{\widehat{\sigma}_{x,y}}{\sqrt{n}} z_{1-\frac{\alpha}{2}}, \, \widehat{q}_{R,n}(y|\mathrm{do}(x)) + \frac{\widehat{\sigma}_{x,y}}{\sqrt{n}} z_{1-\frac{\alpha}{2}} \right], \tag{13}$$

where $z_\beta$ denotes the $\beta$-quantile of a standard normal distribution.

Both $\widehat{q}_{R,n}(y|\mathrm{do}(x))$ and the bounds of the confidence interval can lie outside $[0, 1]$ because the data-generating process adds further constraints on the combination of $P(y|E, x)$, $P(W|E, x)$, and $Q(W)$ that are not reflected by the estimator. To reduce bias and variance, we thus clip the estimate $\widehat{q}_{R,n}(y|\mathrm{do}(x))$ and the confidence intervals to $[0, 1]$.

## 5 Simulation experiments

### 5.1 Data generation

The data generating mechanism can be described using the matrices $P(U|E)$, $Q(U)$, $P(W|U)$, $P(X|U)$ and $\{P(y|U, W, x)\}_{x \in \mathrm{supp}(X), y \in \mathrm{supp}(Y)}$. We randomly and independently generate $M \in \mathbb{N}$ different sets of such matrices. For each choice, we generate $N \in \mathbb{N}$ i.i.d. data sets with sample size $n \in \mathbb{N}$. Appx. C.1 provides details on how we choose the matrices and the size of the causal effects.

### 5.2 Point estimation

We first compare the point estimates obtained from the two proposed estimators. To do so, we fix values $x \in \mathrm{supp}(X)$ and $y \in \mathrm{supp}(Y)$ and compute, for several data sets, the absolute estimation errors $|\widehat{q}_{R,n}(y|\mathrm{do}(x)) - q(y|\mathrm{do}(x))|$ and $|\widehat{q}_{C,n}(y|\mathrm{do}(x)) - q(y|\mathrm{do}(x))|$.

In our setting, we can assess Asm. 1 by considering the condition number $\kappa(P(W|E, x))$. When $\kappa(P(W|E, x))$ is large, the rows of $P(W|E, x)$ are almost linearly dependent, indicating an almost-violation of Asm. 1. By Eq. (4), we thus expect that the performance of $\widehat{q}_{C,n}(y|\mathrm{do}(x))$ and $\widehat{q}_{R,n}(y|\mathrm{do}(x))$ is sensitive to large conditioning numbers and that for larger conditioning numbers, we require larger sample sizes. We validate this empirically in Fig. 3. It shows that the absolute estimation errors of both estimators grow with $\kappa(P(W|E, x))$. It also shows that $\kappa(P(W|E, x)) \approx \widehat{\kappa}(P(W|E, x))$ so we can estimate $\kappa$ directly from the data. Whenever $\widehat{\kappa}(P(W|E, x))$ is large, we can flag the corresponding estimates as potentially unreliable.

The results suggest that both estimation procedures perform similarly in terms of absolute estimation error, with an average value equal to 0.040 for the causal estimator and 0.058 for the reduced estimator, with the reduced estimator being faster to compute (see Appx. C.5 for a runtime analysis). Figs. 7 and 8 in Appx. C.3 show the same results for $n = 1000$ and $n = 100\,000$ supporting the theoretical consistency result. If we add another column to $P(W|E, x)$, that is, another source domain, the reduced estimator seems to perform slightly better than the causal estimator, see Fig. 9 in Appx. C.3.

We further compare the two estimators against other baseline estimators. In particular, we consider an oracle which estimates the causal effect from $n$ i.i.d. realizations of the target intervention

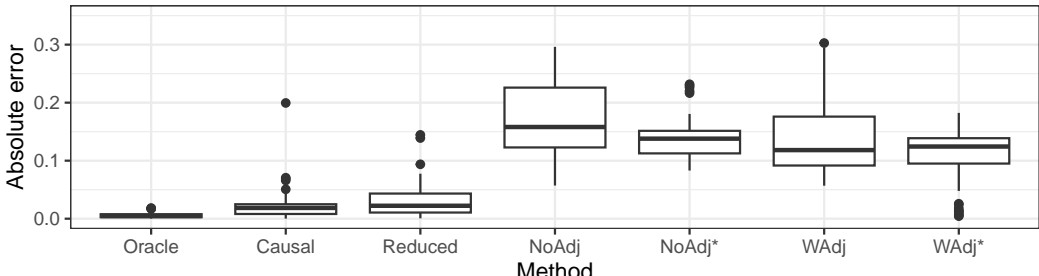

Figure 4: **Comparison of the different estimation procedures.** We use the parameters $k_E = 3$, $n = 20\,000$, $M = 10$, and $N = 5$. The oracle presents the lowest error due to the use of the data from the intervention distribution. The reduced and causal parametrisation estimators show a similar distribution of the error, with their medians close to zero. Their performance is better than that of other baseline methods, whose absolute estimation error distribution is shifted towards larger values. The estimators that use the distribution of $(X, Y)$ in the target domain (with a $^*$) obtain better results in terms of the absolute estimation error than their pooled counterparts.

distribution (using data not available to other methods); the no-adjustment estimator, which estimates the distribution $\mathbb{Q}_Y^{\mathcal{C},\mathrm{do}(x)}$ by $\mathbb{P}_{Y|x}^{\mathcal{C}}$ (NoAdj) or $\mathbb{Q}_{Y|x}^{\mathcal{C}}$ (NoAdj$^*$; using data not available to other methods); and the $W$-adjustment estimator, which uses the adjustment formula in Eq. (4) but with $W$ instead of $U$ (even though $W$ is not a valid adjustment set) using either pooled data from the source domains (WAdj) or from the target domain (WAdj$^*$; using data not available to other methods). The formal definitions of these estimators can be found in Appx. C.3. Fig. 4 shows the comparison in terms of absolute error. The causal and reduced estimators outperform the baselines different from the oracle, which has the lowest error.

## 5.3 Coverage of confidence intervals

Prop. 5 proves asymptotic normality of (the unclipped version of) $\widehat{q}_{R,n}(y|\mathrm{do}(x))$ and provides asymptotically valid confidence intervals. We show that the empirical coverage is close to the nominal value and that the sample median confidence interval length, as suggested by consistency, approaches zero with growing sample size (see Appx. C.4 for details). For comparison, we have also implemented bootstrap confidence intervals based on the normal approximation method [49], see also Appx. C.4 for details. Fig. 10 shows that the empirical coverage of the bootstrap confidence intervals is slightly closer to the nominal level but comes at the cost of longer confidence intervals.

## 6 Application: Hotel searches

We consider the Expedia Hotel Searches dataset [50], in which each observation corresponds to a query made by a user of Expedia's webpage. It contains information about the user, the search filters, the hotels displayed, and whether the user clicked on or booked any of the shown accommodations. We are interested in studying the causal effect of the position of the hotel in the list shown to the user ($X$) on whether the user clicked on the hotel to obtain more information ($Y$). We regard it as plausible that there is an unmeasured confounder $U$ of $X$ and $Y$ (e.g., features of the hotel, such as the distance to the city centre or the additional services provided), but that some of this information is reflected in the price ($W$), which we treat as a proxy. Domains $E$ are comprised of searches in which a given hotel appears. Tab. 1 provides an overview of the involved variables and their role relative to the studied setting. As source domains, we use those hotels that have at least 2000 observations (except for the ones chosen as target domains, see below); this results in 25 hotels. If several of such hotels appear in the same query, this yields several observations (introducing a small dependence between the observations).

The dataset can be viewed as containing both observational and interventional data. For some searches (the ones we use for our estimators), hotels are sorted according to an existing algorithm (unknown to us). For all other searches, hotels are ranked uniformly at random. The latter part allows us to compute a 'ground truth' or 'oracle' causal effect (containing a point estimate and Wald confidence interval) that we can compare against. Ursu [51] uses this dataset to study the effect of

Table 1: **Variables included in the hotel ranking real-world application.**

| Variable | Description | Values |
|---|---|---|
| $E$ | Hotel unique ID | Integers |
| $X$ | Position of the hotel in the search results | $\{1, \ldots, 40\}$ |
| $Y$ | Whether the user clicks on the hotel | 0: No click; 1: Click |
| $W$ | Price range per night in USD | $\{[0, 75], (75, 125], (125, 175], (175, 225], (225, \infty)\}$ |

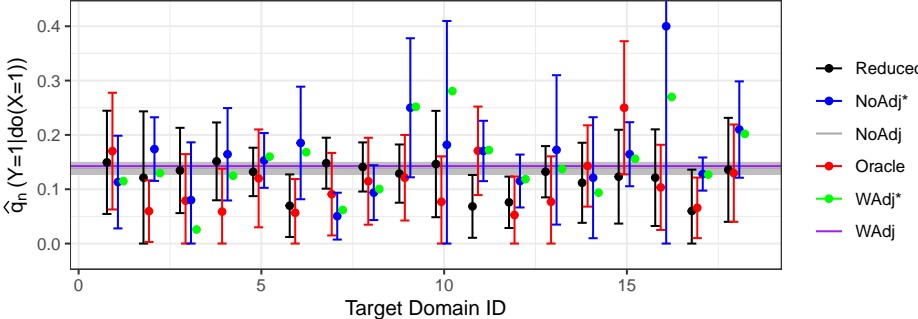

Figure 5: **Reduced parametrisation estimator compared with four baselines and the ground truth for a real dataset.** We compare the estimates of the causal effect $q(Y = 1|\mathrm{do}(X = 1))$ using the reduced estimator and the no-adjustment baselines with the oracle confidence intervals using 25 source and 18 target domains. For both of them, there is overlap between their confidence intervals and the ones from the oracle in all the target domains. The reduced estimator yields estimates closer to the oracle in more target domains and slightly smaller confidence intervals than NoAdj*.

the ranking in the customer choice, but only considers the part of the dataset in which hotels are ranked uniformly at random. This is different from our setting, since we estimate the causal effect $q(Y = 1|\mathrm{do}(X = x))$ in a target domain (the hotel of interest) from the observational data.

In the first experiment, we estimate $q(Y = 1|\mathrm{do}(X = 1))$. As target domains, we choose 18 hotels among those hotels that appear in at least 1500 different queries in the randomized dataset. This ensures that we can obtain reasonable estimates using the oracle, which we consider the ground truth in this experiment. In total, we obtain ca. $64\,000$ and $50\,000$ observations from 8400 and 4000 queries for the source and target domains, respectively. Fig. 5 compares the confidence intervals obtained by the reduced estimator with the oracle ones. As a comparison we also show the result of the no-adjustment and W-adjustment baselines (using both versions: pooled and target), see Appx. C.3, including Wald confidence intervals for the no-adjustment estimator (all confidence intervals in this experiment are at level $0.95$). Generally, both estimators overlap with the oracle confidence intervals. The reduced estimator yields point estimates that are closer to the oracle estimates (average absolute error of 0.044 (Reduced), 0.051 (NoAdj), 0.080 (NoAdj*), 0.053 (WAdj) and 0.075 (WAdj*)) and confidence intervals that are on average shorter than the ones from the no-adjustment baseline (median length of 0.14 versus 0.17). In a second experiment, we estimate the causal effect $q(Y = 1|\mathrm{do}(X = x))$ for different values of $x \in \{1, \ldots, 20\}$ in a fixed target domain to study the causal effect of the different positions on the clicks of the users. Again, the confidence intervals produced by the reduced estimator overlap with all oracle ones, see Appx. D for details.

## 7 Summary and future work

We consider the causal domain adaptation problem of estimating a confounded causal effect in an unseen target domain using data from source domains and an observed proxy variable. We provide two estimators, prove consistency and, in one case, provide asymptotically valid confidence intervals. Experiments on simulated and real data show that these estimators outperform existing methods and support the theoretical claims. We consider the case of a continuous $U$ as an interesting setting, too; here, the ideas discussed by Miao et al. [13] and Tsai et al. [35] could prove helpful.

## Acknowledgments

The project that gave rise to these results received the support of a fellowship from "la Caixa" Foundation (ID 100010434). The fellowship code is LCF/BQ/EU23/12010098.

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

# Appendices

## Table of Contents

# A   Proofs

## A.1   Proof of Thm. 1

**Theorem 1.** *Under the data generating process described in § 2 (ignoring the paragraph "Generalization") and assuming Asm. 1, we have for all $x \in \mathrm{supp}(X)$ and $y \in \mathrm{supp}(Y)$:*

$$q(y|\mathrm{do}(x)) = P(y|E,x)\, P(W|E,x)^{\dagger}\, Q(W). \tag{4}$$

*Therefore, if $(E, W, X, Y)$ is observed in the source domains and $W$ in the target domain, the causal effect of $X$ on $Y$ in the target domain $e_T$ is identifiable.*

*Proof.* The global Markov condition holds in SCMs [e.g., 43, 45], so we have

$$X, Y, W \perp\!\!\!\perp E \mid U \qquad\qquad X \perp\!\!\!\perp W \mid U.$$

We then have

$$P(y|E,x) = P(y|U,x)\, P(U|E,x) \tag{14}$$
$$P(W|E,x) = P(W|U)\, P(U|E,x). \tag{15}$$

For every pair of matrices $A \in \mathbb{R}^{m \times n}$ and $B \in \mathbb{R}^{n \times p}$, the following inequality holds: $\mathrm{rank}(AB) \leq \min\{\mathrm{rank}(A), \mathrm{rank}(B)\}$ [52]. Using this inequality in Eq. (15), we get

$$\mathrm{rank}(P(W|U)) \geq \mathrm{rank}(P(W|E,x)) = k_W,$$

where the last equality is due to Prop. 6. The matrix $P(W|U)$ has dimensions $k_W \times k_U$, hence

$$\min\{k_W, k_U\} \geq \mathrm{rank}(P(W|U)) \geq k_W. \tag{16}$$

Using Eq. (16) and the inequality $k_W \geq k_U$ derived from Asm. 1, we conclude that $\mathrm{rank}(P(W|U)) = k_W = k_U$. Therefore, $P(W|U)$ is an invertible matrix and this can be used in Eq. (15) to obtain

$$P(U|E,x) = P(W|U)^{-1}\, P(W|E,x).$$

The substitution of this expression in Eq. (14) yields

$$P(y|E,x) = P(y|U,x)\, P(W|U)^{-1}\, P(W|E,x),$$

and therefore

$$P(y|U,x) = P(y|E,x)\, P(W|E,x)^{\dagger}\, P(W|U). \tag{17}$$

The adjustment formula (3) states that $q(y|\mathrm{do}(x)) = Q(y|U,x)\, Q(U)$ and the equality $Q(y|U,x) = P(y|U,x)$ holds because of modularity or invariance [e.g., 53]. Combining Eqs. (3) and (17), we get

$$q(y|\mathrm{do}(x)) = P(y|U,x)\, Q(U) = P(y|E,x)\, P(W|E,x)^{\dagger}\, P(W|U)\, Q(U).$$

Using the law of total probability for the last two terms, we get our final result

$$q(y|\mathrm{do}(x)) = P(y|E,x)\, P(W|E,x)^{\dagger}\, Q(W). \tag{18}$$

$\square$

## A.2   Proof of Prop. 3

**Proposition 3.** *For all $x \in \mathrm{supp}(X)$ and $y \in \mathrm{supp}(Y)$ we have[7]*

$$q(y|\mathrm{do}(x)) = \mathrm{diag}\Big(P(y|U,W,x)\, P(W|U)\Big)\, Q(U). \tag{5}$$

*Proof.* By modularity, we have $Q(y|U,x) = P(y|U,x)$. The statement then follows by the covariate adjustment formula (3):

$$q(y|\mathrm{do}(x)) = P(y|U,x)\, Q(U) = \sum_{r=1}^{k_U} p(y|u_r,x)\, q(u_r)$$

$$= \sum_{r=1}^{k_U} \left( \sum_{j=1}^{k_W} p(y|u_r, w_j, x)\, p(w_j|u_r) \right) q(u_r) = \mathrm{diag}\Big(P(y|U,W,x)\, P(W|U)\Big)\, Q(U).$$

$\square$

---

[7]The diag operator applied to a square matrix $A = (A_{i,j})_{1 \leq i,j \leq n} \in \mathbb{R}^{n \times n}$ is defined as $\mathrm{diag}(A) := (A_{1,1}, A_{2,2}, \dots, A_{n,n}) \in \mathbb{R}^n$.

### A.3 Proof of Prop. 4

**Proposition 4.** *Suppose that Asm. 1 holds. The estimator $\widehat{q}_{C,n}(y|\mathrm{do}(x))$ is consistent, that is,*

$$\widehat{q}_{C,n}(y|\mathrm{do}(x)) \rightarrow q(y|\mathrm{do}(x)) \text{ in probability as } n_{\mathrm{src}}, n_{\mathrm{tgt}} \rightarrow \infty.$$

*Proof.* We consider a parameter whose components are the joint pmfs that appear in the likelihood function in Eq. (7)

$$\mu := \big(p_\mu(y_1, x_1, w_1|e_1), \dots, p_\mu(y_{k_Y}, x_{k_X}, w_{k_W}|e_{k_E}), q_\mu(w_1), \dots, q_\mu(w_{k_W})\big)$$

and denote by $\mu_0$ its true value. Moreover, we denote by $f(\widehat{\theta}_n)$ the vector obtained when calculating these joint pmfs from the optimal vector $\widehat{\theta}_n$. Hence,

$$f(\widehat{\theta}_n) = \big(p_{\widehat{\theta}_n}(y_1, x_1, w_1|e_1), \dots, p_{\widehat{\theta}_n}(y_{k_Y}, x_{k_X}, w_{k_W}|e_{k_E}), q_{\widehat{\theta}_n}(w_1), \dots, q_{\widehat{\theta}_n}(w_{k_W})\big).$$

The conditional log-likelihood equals

$$\ell_n(\mu) := \sum_{i=1}^{k_Y}\sum_{s=1}^{k_X}\sum_{j=1}^{k_W}\sum_{l=1}^{k_E} n(y_i, x_s, w_j, e_l)\log(p_\mu(y_i, x_s, w_j|e_l)) + \sum_{j=1}^{k_W} n(w_j)\log(q_\mu(w_j)),$$

where we define $0 \cdot \log 0 = 0$. (It suffices to consider the conditional log-likelihood, as it differs from the unconditional log-likelihood only by an additive term that models the marginal probabilities of the domains.) Instead of $\ell_n$, we now consider

$$\tilde{\ell}_n(\mu) := \sum_{i=1}^{k_Y}\sum_{s=1}^{k_X}\sum_{j=1}^{k_W}\sum_{l=1}^{k_E} \frac{n(y_i, x_s, w_j, e_l)}{n(e_l)}\log(p_\mu(y_i, x_s, w_j|e_l)) + \sum_{j=1}^{k_W} \frac{n(w_j)}{n_{\mathrm{tgt}}}\log(q_\mu(w_j)),$$

which satisfies $\operatorname{argmax}\ell_n = \operatorname{argmax}\tilde{\ell}_n$ (this is because $\ell_n$ can be optimized by considering the likelihood for each domain separately, so multiplying each domain likelihood by a constant $1/n(e_l)$ or $1/n_{\mathrm{tgt}}$ does not change the optimal values for the probabilities). The unique value for $\mu$ maximizing both $\tilde{\ell}_n$ and $\ell_n$, denoted by $\widehat{\mu}_n$, consists of the relative frequencies

$$p_{\widehat{\mu}_n}(y_i, x_s, w_j|e_l) = \frac{n(y_i, x_s, w_j, e_l)}{n(e_l)},$$

$$q_{\widehat{\mu}_n}(w_j) = \frac{n(w_j)}{n_{\mathrm{tgt}}}.$$

For all $n \in \mathbb{N}$,

$$\ell_n(\widehat{\mu}_n) \geq \ell_n(f(\widehat{\theta}_n)) \geq \ell_n(\mu_0). \tag{19}$$

The first inequality is due to the definition $\widehat{\mu}_n := \operatorname{argmax}\ell_n$. Regarding the second inequality, we have

$$\widehat{\theta}_n := \operatorname{argmax} L = \operatorname{argmax}\ \log L = \operatorname{argmax}\ \ell_n \circ f,$$

(where the last equality is due to $\log L = \ell_n \circ f$, which holds by definition of $f$); and since there exists $\theta_0$ such that $\mu_0 = f(\theta_0)$, we have

$$\ell_n(f(\widehat{\theta}_n)) \geq \ell_n(f(\theta_0)) = \ell_n(\mu_0).$$

We have that $\widehat{\mu}_n \xrightarrow{P} \mu_0$ due to the law of large numbers and we use the continuous mapping theorem to get

$$\frac{\ell_n(\widehat{\mu}_n) - \ell_n(\mu_0)}{n_{\min}} \xrightarrow{P} 0,$$

where $n_{\min} := \min\{n(e_1), \dots, n(e_{k_E}), n_{\mathrm{tgt}}\}$ is the smallest sample size across all source and target domains. Using the squeeze theorem on Eq. (19),

$$\frac{\ell_n(\widehat{\mu}_n) - \ell_n(f(\widehat{\theta}_n))}{n_{\min}} \xrightarrow{P} 0.$$

Furthermore, we have the inequality

$$\frac{\ell_n(\widehat{\mu}_n) - \ell_n(f(\widehat{\theta}_n))}{n_{\min}} \geq \tilde{\ell}_n(\widehat{\mu}_n) - \tilde{\ell}_n(f(\widehat{\theta}_n)).$$

To prove this inequality, we consider the decomposition

$$\ell_n(\mu) =: \sum_{l=1}^{k_E} \ell_n^{e_l}(\mu) + \ell_n^{e_T}(\mu)$$

corresponding to the sum of the likelihoods in each domain. For all $l \in \{1, \ldots, k_E, T\}$, the vector $\widehat{\mu}_n$ is the maximizer of $\ell_n^{e_l}$, so we have

$$\ell_n^{e_l}(\widehat{\mu}_n) - \ell_n^{e_l}(f(\widehat{\theta}_n)) \geq 0. \tag{20}$$

Therefore,

$$\tilde{\ell}_n(\widehat{\mu}_n) - \tilde{\ell}_n(f(\widehat{\theta}_n))$$

$$= \sum_{i=1}^{k_Y} \sum_{s=1}^{k_X} \sum_{j=1}^{k_W} \sum_{l=1}^{k_E} \frac{n(y_i, x_s, w_j, e_l)}{n(e_l)} \log\left(\frac{p_{\widehat{\mu}_n}(y_i, x_s, w_j | e_l)}{p_{\widehat{\theta}_n}(y_i, x_s, w_j | e_l)}\right) + \sum_{j=1}^{k_W} \frac{n(w_j)}{n_{\text{tgt}}} \log\left(\frac{q_{\widehat{\mu}_n}(w_j)}{q_{\widehat{\theta}_n}(w_j)}\right)$$

$$= \sum_{l=1}^{k_E} \frac{1}{n(e_l)} \left(\ell_n^{e_l}(\widehat{\mu}_n) - \ell_n^{e_l}(f(\widehat{\theta}_n))\right) + \frac{1}{n_{\text{tgt}}} \left(\ell_n^{e_T}(\widehat{\mu}_n) - \ell_n^{e_T}(f(\widehat{\theta}_n))\right)$$

$$\leq \frac{1}{n_{\min}} \left(\sum_{l=1}^{k_E} \left(\ell_n^{e_l}(\widehat{\mu}_n) - \ell_n^{e_l}(f(\widehat{\theta}_n))\right) + \left(\ell_n^{e_T}(\widehat{\mu}_n) - \ell_n^{e_T}(f(\widehat{\theta}_n))\right)\right)$$

$$= \frac{\ell_n(\widehat{\mu}_n) - \ell_n(f(\widehat{\theta}_n))}{n_{\min}},$$

where the inequality holds due to Eq. (20). Therefore, considering the Kullback-Leibler divergence, we get the following convergence in probability[8]:

$$\sum_{l=1}^{k_E} D_{\text{KL}}\left(\widehat{\mu}_n^{e_l} || f(\widehat{\theta}_n)^{e_l}\right) + D_{\text{KL}}\left(\widehat{\mu}_n^{e_T} || f(\widehat{\theta}_n)^{e_T}\right) = \tilde{\ell}_n(\widehat{\mu}_n) - \tilde{\ell}_n(f(\widehat{\theta}_n)) \xrightarrow{P} 0,$$

which implies that each of the summands on the left-hand side converges to zero in probability. Then, by Pinsker's inequality and the continuous mapping theorem, we have

$$\|\widehat{\mu}_n - f(\widehat{\theta}_n)\|_1 \leq \sum_{l=1}^{k_E} \sqrt{2 D_{\text{KL}}\left(\widehat{\mu}_n^{e_l} || f(\widehat{\theta}_n)^{e_l}\right)} + \sqrt{2 D_{\text{KL}}\left(\widehat{\mu}_n^{e_T} || f(\widehat{\theta}_n)^{e_T}\right)} \xrightarrow{P} 0.$$

Hence,

$$f(\widehat{\theta}_n) \xrightarrow{P} \mu_0.$$

Finally, we apply the continuous mapping theorem with the function $\tilde{g}_{x,y} : \mu_0 \longmapsto \tilde{g}_{x,y}(\mu_0) := q(y|\text{do}(x))$ (defined, for example, using Eq. (4), where all the components can be derived from the joint pmfs in $\mu_0$) to get the final result:

$$\widehat{q}_{C,n}(y|\text{do}(x)) \xrightarrow{P} q(y|\text{do}(x)).$$

$\square$

## A.4 Proof of Prop. 5

**Proposition 5.** *Suppose that Asm. 1 holds. For all $x \in \text{supp}(X)$ and $y \in \text{supp}(Y)$, the estimator $\widehat{q}_{R,n}(y|\text{do}(x))$ defined in Eq. (12) is a consistent estimator of $q(y|\text{do}(x))$. Furthermore,*

$$\frac{\sqrt{n}}{\widehat{\sigma}_{x,y}} \left(\widehat{q}_{R,n}(y|\text{do}(x)) - q(y|\text{do}(x))\right) \xrightarrow{\mathcal{D}} \mathcal{N}(0, 1),$$

*where*

$$\widehat{\sigma}_{x,y}^2 := \nabla h(\widehat{\eta}_{x,y,n})^\top \widehat{\Sigma}_{x,y} \nabla h(\widehat{\eta}_{x,y,n}),$$

*with $\widehat{\Sigma}_{x,y}$ being the sample covariance matrix of $(\eta_{x,y}^1, \ldots, \eta_{x,y}^n)$.*

[8] Here $\widehat{\mu}_n^{e_1} := (p_{\widehat{\mu}_n}(y_1, x_1, w_1 | e_1), \ldots, p_{\widehat{\mu}_n}(y_{k_Y}, x_{k_X}, w_{k_W} | e_1))$ and analogously for the other variables.

*Proof.* Consider the i.i.d. sample $\{\eta_{x,y}^i\}_{i=1}^n$ obtained from $\{(E_i, W_i, X_i, Y_i)\}_{i=1}^n$ using the definition in Eq. (9). The random vector $\eta_{x,y}^i$ has mean equal to the true parameter $\eta_{x,y,0}$. We define $\widehat{\eta}_{x,y,n}^*$ to be the sample mean of $\eta_{x,y}$. Here, we use the asterisk to stress the difference to the estimator $\widehat{\eta}_{x,y,n}$, which is equal to $\widehat{\eta}_{x,y,n}^*$ except for when $P_{\widehat{\eta}_{x,y,n}^*}(W|E, x)$ does not have linearly independent rows.

To prove asymptotic normality, we apply the central limit theorem to $\{\eta_{x,y}^i\}_{i=1}^n$. We use again that their mean is equal to $\eta_{x,y,0}$ and their sample mean is the estimator $\widehat{\eta}_{x,y,n}^*$. Furthermore, we denote by $\Sigma_{x,y}$ the covariance matrix of $\eta_{x,y}^1$, obtaining

$$\sqrt{n}\Big(\widehat{\eta}_{x,y,n}^* - \eta_{x,y,0}\Big) \xrightarrow{\mathcal{D}} \mathcal{N}(0, \Sigma_{x,y}). \tag{21}$$

We now prove that

$$\sqrt{n}\Big(\widehat{\eta}_{x,y,n} - \eta_{x,y,0}\Big) \xrightarrow{\mathcal{D}} \mathcal{N}(0, \Sigma_{x,y}) \tag{22}$$

holds, too. Let $B_n$ be the event that $P_{\widehat{\eta}_{x,y,n}^*}(W|E, x)$ has linearly independent rows. We first show that $\mathbb{P}(B_n) \to 1$. Indeed, by Asm. 1 and Prop. 6, the population matrix $P(W|E, x)$ has full row rank. Thus, there is a $k_W \times k_W$ submatrix $M$ s.t. $\det(P(W|E, x)_M) = c \neq 0$. By continuity of the determinant, there is a $\delta > 0$ s.t.[9] $\|P(W|E, x)_M - P_{\widehat{\eta}_{x,y,n}^*}(W|E, x)_M\|_F < \delta$ (let us call such events $A_{n,\delta}$) implies $\det(P_{\widehat{\eta}_{x,y,n}^*}(W|E, x)_M) \neq 0$, so $P_{\widehat{\eta}_{x,y,n}^*}(W|E, x)$ has full row rank. By the law of large numbers, $\mathbb{P}(A_{n,\delta}) \to 1$ and thus $\mathbb{P}(B_n) \to 1$. Defining $\hat{a}_n := \sqrt{n}(\widehat{\eta}_{x,y,n} - \eta_{x,y,0})$ and $\hat{a}_n^* := \sqrt{n}(\widehat{\eta}_{x,y,n}^* - \eta_{x,y,0})$, we can now prove $\hat{a}_n \xrightarrow{\mathcal{D}} \mathcal{N}(0, \Sigma_{x,y})$. Indeed, for all $\mathbf{b}$, we have

$$\lim_{n\to\infty} \mathbb{P}(\hat{a}_n \leq \mathbf{b}) = \lim_{n\to\infty} \big(\mathbb{P}(\hat{a}_n \leq \mathbf{b} \cap B_n) + \mathbb{P}(\hat{a}_n \leq \mathbf{b} \cap B_n^C)\big)$$

$$= \lim_{n\to\infty} \big(\mathbb{P}(\hat{a}_n^* \leq \mathbf{b} \cap B_n) + \mathbb{P}(\hat{a}_n \leq \mathbf{b} \cap B_n^C) + \mathbb{P}(\hat{a}_n^* \leq \mathbf{b} \cap B_n^C)\big)$$

$$= \lim_{n\to\infty} \big(\mathbb{P}(\hat{a}_n^* \leq \mathbf{b}) + \mathbb{P}(\hat{a}_n \leq \mathbf{b} \cap B_n^C)\big)$$

$$= \lim_{n\to\infty} \mathbb{P}(\hat{a}_n^* \leq \mathbf{b}),$$

which proves Eq. (22). The second equality holds because, on $B_n$, $\hat{a}_n^*$ and $\hat{a}_n$ are identical and $\lim_{n\to\infty} \mathbb{P}(\hat{a}_n^* \leq \mathbf{b} \cap B_n^C) = 0$.

The function $h$ defined in Eq. (11) is continuous, so we apply the delta method to Eq. (22) to get

$$\sqrt{n}\Big(\widehat{q}_{R,n}(y|\mathrm{do}(x)) - q(y|\mathrm{do}(x))\Big) \xrightarrow{\mathcal{D}} \mathcal{N}(0, \sigma_{x,y}^2),$$

where the variance $\sigma_{x,y}^2$ is given by

$$\sigma_{x,y}^2 = \nabla h(\eta_{x,y,0})^\top \Sigma_{x,y} \nabla h(\eta_{x,y,0}).$$

Consistency follows by Slutsky's theorem.

To estimate $\sigma_{x,y}^2$, we can replace $\Sigma_{x,y}$ by the sample covariance matrix $\widehat{\Sigma}_{x,y,n}$ of $(\eta_{x,y}^1, \ldots, \eta_{x,y}^n)$ and $\eta_{x,y,0}$ by $\widehat{\eta}_{x,y,n}$, which are both consistent estimators by the law of large numbers. This yields

$$\widehat{\sigma}_{x,y}^2 := \nabla h(\widehat{\eta}_{x,y,n})^\top \widehat{\Sigma}_{x,y,n} \nabla h(\widehat{\eta}_{x,y,n}).$$

By the continuous mapping theorem, $\widehat{\sigma}_{x,y}^2$ is a consistent estimator of $\sigma_{x,y}^2$. Therefore, using the continuous mapping theorem again, together with Slutsky's theorem, we have

$$\frac{\sqrt{n}}{\widehat{\sigma}_{x,y}}\Big(\widehat{q}_{R,n}(y|\mathrm{do}(x)) - q(y|\mathrm{do}(x))\Big) \xrightarrow{\mathcal{D}} \mathcal{N}(0, 1).$$

$\square$

# B   Details and extensions of theoretical results

In this appendix, we provide several additional results that highlight how the assumptions and the setting studied in the main paper can be relaxed.

---

[9] $\|\cdot\|_F$ denotes the Frobenius norm.

## B.1 Relaxing Asm. 1

Under Asm. 1, the proxy $W$ can be transformed in such a way that the matrix $P(W|E, x)$ has linearly independent rows.

**Proposition 6.** *Under Asm. 1, for all $x \in \operatorname{supp}(X)$, there exists a $W$-measurable random variable $\tilde{W}$ such that*

$$k_{\tilde{W}} = \operatorname{rank}(P(\tilde{W}|E, x)) = \operatorname{rank}(P(W|E, x))$$

*and $\tilde{W}$ satisfies the independence statements $(X, Y, \tilde{W} \perp\!\!\!\perp E \,|\, U)$ and $(X \perp\!\!\!\perp \tilde{W} \,|\, U)$, and $\operatorname{supp}((E, U, \tilde{W}, X, Y)) = \operatorname{supp}(E) \times \operatorname{supp}(U) \times \operatorname{supp}(\tilde{W}) \times \operatorname{supp}(X) \times \operatorname{supp}(Y)$.*

**Lemma 7.** *Let $w_1, \ldots, w_n$ be linearly independent vectors in a vector space $V$, and let*

$$v = \sum_{i=1}^{n} \lambda_i \, w_i,$$

*with scalars $\lambda_i \in \mathbb{R}$. If $\lambda_1 \neq -1$, then*

$$v + w_1, w_2, \ldots, w_n$$

*are linearly independent.*

*Proof.* Write

$$v + w_1 = (\lambda_1 + 1) \, w_1 + \sum_{i=2}^{n} \lambda_i \, w_i.$$

Suppose there exist scalars $a, b_2, \ldots, b_n$ such that

$$a\,(v + w_1) \; + \; \sum_{i=2}^{n} b_i \, w_i \; = \; 0.$$

Substituting the expansion of $v + w_1$ gives

$$a \Big[ (\lambda_1 + 1) w_1 + \lambda_2 w_2 + \cdots + \lambda_n w_n \Big] \; + \; \sum_{i=2}^{n} b_i \, w_i \; = \; 0.$$

Collecting coefficients on the independent set $\{w_1, \ldots, w_n\}$ yields the system

$$\begin{cases} a\,(\lambda_1 + 1) = 0, \\ a\,\lambda_i + b_i = 0, \quad i \in \{2, \ldots, n\}. \end{cases}$$

Since $\lambda_1 + 1 \neq 0$, the first equation forces $a = 0$, which implies $b_2 = \ldots = b_n = 0$. Hence, the only solution is the trivial one, proving that

$$v + w_1, w_2, \ldots, w_n$$

are linearly independent. $\qquad\square$

*Proof of Prop. 6.* Fix $x \in \operatorname{supp}(X)$. If $\operatorname{rank}(P(W|E, x)) = k_W$ the result follows considering $\tilde{W} = W$. Assume $r := \operatorname{rank}(P(W|E, x)) < k_W$. We proceed as follows: Choose $r + 1$ rows of $P(W|E, x)$, denoted by $v_1, \ldots, v_{r+1}$ (with row indices denoted by $a_1, \ldots, a_{r+1}$), such that $v_1, \ldots, v_{i-1}, v_{i+1}, \ldots, v_{r+1}$ are linearly independent and $v_i$ is a linear combination of the others, that is, there exist $\lambda_1, \ldots, \lambda_{i-1}, \lambda_{i+1}, \ldots, \lambda_{r+1} \in \mathbb{R}$ such that $v_i = \sum_{j \neq i} \lambda_j v_j$. Since the entries of $P(W|E, x)$ are non-negative, there exists $k \in \{1, \ldots, r + 1\} \setminus \{i\}$ such that $\lambda_k \neq -1$. We define a new random variable $\tilde{W} := W \mathbb{1}\{W \neq w_{a_i}\} + w_{a_k} \mathbb{1}\{W = w_{a_i}\}$. By Lemma 7 we have that $P(\tilde{W}|E, x)$ is a $(k_W - 1) \times k_E$-dimensional matrix and $\operatorname{rank}(P(\tilde{W}|E, x)) = \operatorname{rank}(P(W|E, x))$. It also holds that $\operatorname{supp}((E, U, \tilde{W}, X, Y)) = \operatorname{supp}(E) \times \operatorname{supp}(U) \times \operatorname{supp}(\tilde{W}) \times \operatorname{supp}(X) \times \operatorname{supp}(Y)$. Continue this procedure until all rows are independent. By an abuse of notation we call the resulting random variable again $\tilde{W}$. For all random variables $A, B, C$ and $f$ measurable $A \perp\!\!\!\perp B \,|\, C \implies f(A) \perp\!\!\!\perp B \,|\, C$. It therefore holds that

$$X, Y, \tilde{W} \perp\!\!\!\perp E \,|\, U \qquad\qquad X \perp\!\!\!\perp \tilde{W} \,|\, U.$$

We have that $\operatorname{rank}(P(\tilde{W}|E, x)) = \operatorname{rank}(P(W|E, x)) \geq k_U$ and $P(\tilde{W}|E, x)$ has full row rank. $\quad\square$

## B.2  Asm. 1 and identifiability

We now present a counterexample showing that identifiability is, in general, not possible if Asm. 1 is violated. Concretely, we now construct two different SCMs. Both of them satisfy $\mathrm{supp}(X) := \{0, 1\}$ and $\mathrm{supp}(Y) := \{0, 1\}$. We consider $x := 0 \in \mathrm{supp}(X)$ and $y := 0 \in \mathrm{supp}(Y)$ and are interested in inferring $q(y|\mathrm{do}(x))$. Both SCMs satisfy $\mathrm{supp}(W) := \{w_1, w_2, w_3\}$, $\mathrm{supp}(U) := \{u_1, u_2, u_3\}$ and, for both SCMs, $P(W|U)$ and $P(y|U, W, x)$ are such that, for all $j \in \{1, 2, 3\}$,

$$P(W|U) = \begin{pmatrix} 0.23 & 0.3 & 0.2 \\ 0.46 & 0.6 & 0.4 \\ 0.31 & 0.1 & 0.4 \end{pmatrix} \quad \text{and} \quad P(y|U, w_j, x) = P(y|U, x) = (0.5 \quad 0.2 \quad 0.3).$$

Furthermore, we choose, again for both SCMs, $P(E)$, $P(U|E)$ and $P(X|U)$, such that the induced source distribution has full support over $(E, U, W, X, Y)$ (this is possible, as sums of products of strictly positive numbers are strictly positive). Denoting by $c_1$, $c_2$, and $c_3$ the columns of $P(W|U)$, we have

$$c_1 = \frac{3}{10}c_2 + \frac{7}{10}c_3, \tag{23}$$

with $c_2$ and $c_3$ being linearly independent, so $\mathrm{rank}(P(W|U)) = 2 < k_U$. Using Eq. (15) and the inequality $\mathrm{rank}(AB) \leq \min\{\mathrm{rank}(A), \mathrm{rank}(B)\}$ [52], we get that $\mathrm{rank}(P(W|E, x)) < k_U$, so in both SCMs Asm. 1 is violated.

The two SCMs differ in the target distribution of $U$: we define

$$P_1(U) := \begin{pmatrix} p_1(u_1) \\ p_1(u_2) \\ p_1(u_3) \end{pmatrix} = \begin{pmatrix} 0.6 \\ 0.3 \\ 0.1 \end{pmatrix} \quad \text{and} \quad P_2(U) := \begin{pmatrix} p_2(u_1) \\ p_2(u_2) \\ p_2(u_3) \end{pmatrix} = \begin{pmatrix} 0.5 \\ 0.33 \\ 0.17 \end{pmatrix}.$$

The induced target distributions of $W$ for the two SCMs are given by the law of total probability:

$$P_1(W) = P(W|U)P_1(U) = 0.6c_1 + 0.3c_2 + 0.1c_3 = \begin{pmatrix} 0.248 \\ 0.496 \\ 0.256 \end{pmatrix}$$

$$= 0.5c_1 + 0.33c_2 + 0.17c_3 = P(W|U)P_2(U) = P_2(W).$$

Therefore, both SCMs induce the same target distribution of the variable $W$. Thus, both SCMs induce the same source distributions over $(E, X, Y, W)$ and the same target distribution over $W$, and thereby agree on the distributions over all observed quantities.

However, if we calculate the causal effect $q(y|\mathrm{do}(x))$ for each SCM using the covariate adjustment formula (see Eq. (3)), we find that they differ:

$$q_1(y|\mathrm{do}(x)) = \sum_{r=1}^{3} p(y|u_r, x)p_1(u_r) = 0.39$$

$$\neq q_2(y|\mathrm{do}(x)) = \sum_{r=1}^{3} p(y|u_r, x)p_2(u_r) = 0.367.$$

In conclusion, the causal effect from $X$ to $Y$ is not identifiable.

Logically, the counterexample implies that if the causal effect from $X$ to $Y$ is identified via Eq. (4) for all SCMs satisfying all assumptions except for possibly Asm. 1 (implying, in particular, that the pseudo-inverse $P(W|E, x)^\dagger$ exists), then Asm. 1 must hold, too.

## B.3  Reduced parametrisation

The parameter vector $\eta_{x,y}$ should have components corresponding to the following probabilities:

$$\Big( q_{\eta_{x,y}}(w_1, e_T), \ldots, q_{\eta_{x,y}}(w_{k_W-1}, e_T), q_{\eta_{x,y}}(e_T), p_{\eta_{x,y}}(w_1, x, e_1), \ldots,$$

$$p_{\eta_{x,y}}(w_{k_W-1}, x, e_{k_E}), p_{\eta_{x,y}}(y, x, e_1), \ldots, p_{\eta_{x,y}}(y, x, e_{k_E}), p_{\eta_{x,y}}(x, e_1), \ldots, p_{\eta_{x,y}}(x, e_{k_E}) \Big).$$

We denote the number of components of this vector by $k_\eta := k_W + (k_W + 1)k_E$, so the corresponding parameter space is contained in $H = [0, 1]^{k_\eta}$.

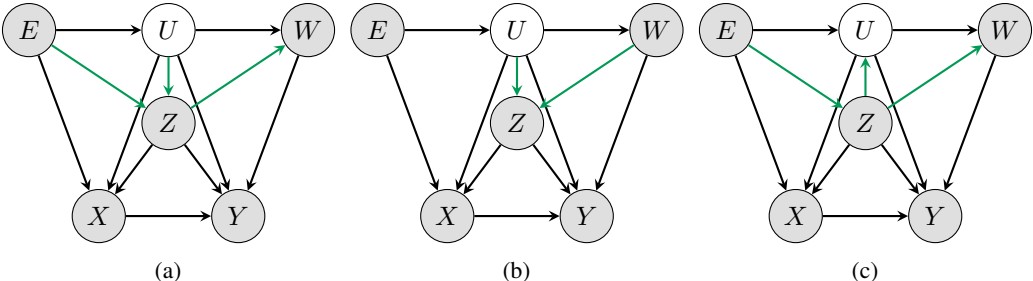

Figure 6: **Causal effect estimation in unseen domains with both observed and unobserved confounders.** Shown are three example graphs that allow for identification of the conditional and unconditional causal effect in the presence of an additional observed confounder, see Thm. 8 for details. The black edges are shared across all graphs; the green edges differ.

## B.4 Additional covariate/observed confounder

We now consider the case in which we have an additional observed confounder $Z$. Specifically, suppose that the SCM induces one of the graphs depicted in Fig. 6.

We replace Asm. 1 with the following assumption.

**Assumption 2.** *The proxy variable $W$ is such that $k_W \geq k_U$ and, for all $x \in \mathrm{supp}(X)$ and $z \in supp(Z)$, the matrix $P(W|E, z, x)$ has linearly independent rows.*

**Theorem 8.** *Assume an SCM whose induced graph is one of those shown in Figure 6. Let $E, U, W, X, Y, Z$ denote the random variables generated by this SCM, and suppose their joint support factorizes as $\mathrm{supp}(E, U, W, X, Y, Z) = \mathrm{supp}(E) \times \mathrm{supp}(U) \times \mathrm{supp}(W) \times \mathrm{supp}(X) \times \mathrm{supp}(Y) \times \mathrm{supp}(Z)$. Suppose that Asm. 2 holds. We then have for all $x \in \mathrm{supp}(X)$, $z \in \mathrm{supp}(Z)$, and $y \in \mathrm{supp}(Y)$:*

$$q(y|\mathrm{do}(x), z) = P(y|E, x, z)P(W|E, x, z)^{\dagger}Q(W|z)$$

*and*

$$q(y|\mathrm{do}(x)) = \sum_{z \in \mathrm{supp}(Z)} P(y|E, x, z)P(W|E, x, z)^{\dagger}Q(W|z)q(z).$$

*Therefore, if $(E, W, X, Y, Z)$ is observed in the source domains and $(W, Z)$ in the target domain, the conditional causal effect of $X$ on $Y$ given $Z = z$ and the total causal effect of $X$ on $Y$ in the target domain $e_T$ are identifiable.*

*Proof.* Since the global Markov condition holds for the induced graph and distribution of SCMs, we have for the graphs in Fig. 6 that $Y \perp\!\!\!\perp E|(U, X, Z)$ and $W \perp\!\!\!\perp (E, X)|(U, Z)$ and thus for all $x \in \mathrm{supp}(X), y \in \mathrm{supp}(Y), z \in \mathrm{supp}(Z)$:

$$P(y|E, x, z) = P(y|U, x, z)P(U|E, x, z)$$
$$P(W|E, x, z) = P(W|U, z)P(U|E, x, z).$$

Under Asm. 2, we follow the same steps as in the proof in Appx. A.1 to get

$$P(y|U, x, z) = P(y|E, x, z)P(W|E, x, z)^{\dagger}P(W|U, z). \tag{24}$$

The target conditional interventional distribution is given by

$$
\begin{aligned}
q(y|\mathrm{do}(x), z) &= Q(y|U, \mathrm{do}(x), z)Q(U|\mathrm{do}(x), z) \\
&= Q(y|U, \mathrm{do}(x), z)Q(U|z) \\
&= Q(y|U, x, z)Q(U|z) \\
&= P(y|U, x, z)Q(U|z),
\end{aligned}
$$

where the first line follows from the law of total probability; the second line follows the fact that $U \perp\!\!\!\perp X|Z$ in the post-intervention graph $\mathcal{G}_{\overline{X}}$ (formally, do-calculus rule 3); the third line follows

from parent adjustment applied to $x$ (formally, do-calculus rule 2); and the last line follows from the assumed modularity or invariance. Substituting the expression from Eq. (24) then yields

$$q(y|\text{do}(x), z) = P(y|E, x, z)P(W|E, x, z)^\dagger P(W|U, z)Q(U|z) \tag{25}$$

$$= P(y|E, x, z)P(W|E, x, z)^\dagger Q(W|z). \tag{26}$$

Hence, if Asm. 2 holds and $(W, Z)$ is observed in the target domain, the conditional interventional distribution $q(y|\text{do}(x), z)$ is identified.

Furthermore,

$$q(y|\text{do}(x)) = \sum_{z \in \text{supp}(Z)} q(y|\text{do}(x), z)q(z|\text{do}(x))$$

$$= \sum_{z \in \text{supp}(Z)} q(y|\text{do}(x), z)q(z)$$

$$= \sum_{z \in \text{supp}(Z)} P(y|E, x, z)P(W|E, x, z)^\dagger Q(W|z)q(z),$$

where the first line follows from the law of total probability; the second from $Z \perp\!\!\!\perp X$ in the post-intervention graph $\mathcal{G}_{\overline{X}}$; and the last by substitution of (26). Therefore, if $(W, X, Y, Z)$ is observed in the source domains and $(W, Z)$ in the target domain, the causal effect $q(y|\text{do}(x))$ is identifiable. $\square$

**Remark 9** (Relaxing the latent shift assumption via $E \to X$). *The existence of an edge from $E$ to $X$ in the graphs in Fig. 6 means that the latent shift assumption required by Alabdulmohsin et al. [33] and Tsai et al. [35] (which we also rely on in the main paper) can be relaxed in our setting. Specifically, we can also allow for (direct or unmediated) covariate shift, i.e.,*

$$X \not\perp\!\!\!\perp E|(U, Z).$$

*This applies not only to Thm. 8 but also (in the form $X \not\perp\!\!\!\perp E|U$) to Thm. 1, i.e., the identifiability result without observed confounders $Z$ presented in the main paper.*

*The reason for this possible relaxation is that Thm. 1 only makes use of*

$$\{Y \perp\!\!\!\perp E|(U, X), W \perp\!\!\!\perp (E, X)|U\} \quad \text{in} \quad \mathcal{G},$$
$$U \perp\!\!\!\perp X \quad \text{in} \quad \mathcal{G}_{\overline{X}},$$
$$Y \perp\!\!\!\perp X|U \quad \text{in} \quad \mathcal{G}_{\underline{X}},$$

*while Thm. 8 only requires the analogous statements given $Z$,*

$$\{Y \perp\!\!\!\perp E|(U, X, Z), W \perp\!\!\!\perp (E, X)|(U, Z)\} \quad \text{in} \quad \mathcal{G},$$
$$U \perp\!\!\!\perp X|Z \quad \text{in} \quad \mathcal{G}_{\overline{X}},$$
$$Y \perp\!\!\!\perp X|(U, Z) \quad \text{in} \quad \mathcal{G}_{\underline{X}},$$

*where $\mathcal{G}_{\overline{X}}$ and $\mathcal{G}_{\underline{X}}$ denote the graphs obtained by respectively removing the edges coming into $X$ and going out of $\overline{X}$ from $\mathcal{G}$. All these relations still hold when including $E \to X$.*

### B.5 Continuous proxy variable

We can extend our results to continuous proxy variables $W$ with similar ideas as the ones used in Appx. B.1. Let $m \in \mathbb{N}$ and $V_1, \ldots, V_m \subseteq \text{supp}(W)$ be pairwise disjoint measurable sets such that $\text{supp}(W) = \bigcup_{i \leq m} V_i$. Define $\tilde{W}_V := \sum_{i \leq m} i\mathbb{1}\{W \in V_i\}$.

**Assumption 3.** *Assume that, for all $x \in \text{supp}(X)$, there exist $m \in \mathbb{N}$ and $V_1, \ldots, V_m \subseteq \text{supp}(W)$ pairwise disjoint measurable sets such that $\text{supp}(W) = \bigcup_{i \leq m} V_i$ and $\text{rank}(P(\tilde{W}_V|E, x)) \geq k_U$.*

**Proposition 10.** *Under Asm. 3, for all $x \in \text{supp}(X)$, we have for the $W$-measurable random variable $\tilde{W}_V$ that, for all $y \in \text{supp}(Y)$,*

$$q(y|\text{do}(x)) = P(y|E, x)\, P(\tilde{W}_V|E, x)^\dagger\, Q(\tilde{W}_V).$$

*Therefore, if $(E, W, X, Y)$ is observed in the source domains and $W$ in the target domain, the causal effect of $X$ on $Y$ in the target domain $e_T$ is identifiable.*

*Proof.* The result follows directly from Thm. 1 with $W = \tilde{W}_V$ and the observation that

$$X, Y, \tilde{W}_V \perp\!\!\!\perp E \,|\, U \qquad X \perp\!\!\!\perp \tilde{W}_V \,|\, U.$$

$\square$

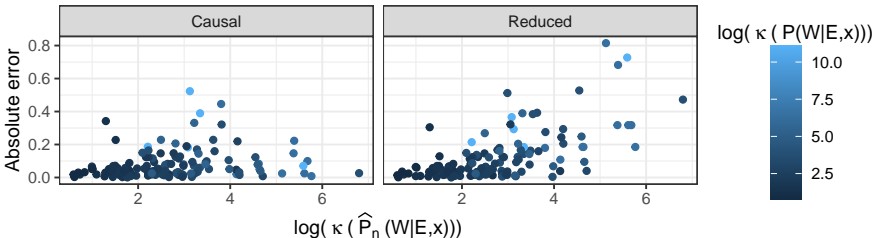

Figure 7: **Comparison of the absolute error for both parametrisations when having a small sample size.** Absolute error for a setting with $k_E = 2$, $n = 1000$, $M = 25$ and $N = 5$. The error is larger in both cases than in Fig. 3, even for the low values of the condition number $\kappa(P(W|E, x))$.

## C   Implementation of the experiments and further simulation results

### C.1   Data generation

The algorithm to generate the data for the simulation studies is based on the SCM introduced in Eq. (1). The algorithm has the following steps:

(i) Generate $P(U|E)$, $Q(U)$, $P(W|U)$, $P(X|U)$ and $\{P(y|U, W, x)\}_{x \in \text{supp}(X), y \in \text{supp}(Y)}$ at random. The sum of each column of $P(U|E)$, $Q(U)$, $P(W|U)$ and $P(X|U)$ has to be equal to 1; we generate each column of length $k$ from a Dirichlet distribution with parameter $\alpha = (1, \ldots, 1) \in \mathbb{R}^k$, so the values are uniformly distributed in the $(k-1)$-dimensional simplex [54]. Similarly, for all $r \in \{1, \ldots, k_U\}$, $j \in \{1, \ldots, k_W\}$ and $s \in \{1, \ldots, k_X\}$, $\sum_{i=1}^n p(y_i|u_r, w_j, x_s) = 1$; we generate the values $p(y_i|u_r, w_j, x_s)$ from a Dirichlet distribution with parameter $(1, \ldots, 1) \in \mathbb{R}^{k_Y}$. The variable $E$ is generated from a discrete uniform distribution in $\{e_1, \ldots, e_{k_E}, e_T\}$, where $e_1, \ldots, e_{k_E}$ are considered the source domains and $e_T$ is the target domain.

(ii) Generate an i.i.d. sample with sample size $n$ of $(E, U, W, X, Y)$ from the induced distribution of the SCM defined by the matrices from step (i) following the assignments in Eq. (1).

(iii) Repeat step (ii) $N$ times.

(iv) Repeat steps (i)–(iii) $M$ times.

This algorithm generates $MN$ samples (each one corresponding to in total $n$ data points, some of which belong to the target domain and the rest to the source domains). The matrices from step (i) are used to calculate the true value $q(y|\text{do}(x))$, which is used to measure the absolute estimation error. Each sample obtained in step (ii) is used to calculate an estimate of the causal effect. The causal effects $q(y|\text{do}(x))$ generated by this algorithm have a sample mean around 0.5.

### C.2   Optimization

The implementation of the causal estimator presented in § 4.1 requires the optimization of the likelihood function in Eq. (7). The components of $\theta$ are optimized first over $\mathbb{R}$ and later transformed into valid pmfs using the softmax function. The optimization is done using the default `optim` function in R with the L-BFGS-B algorithm [55]. The initial value for each component of $\theta$ is generated from a continuous uniform distribution in $[0, 1]$ and we consider a maximum of 50 000 iterations.

### C.3   Point estimation

The plot presented in Fig. 3 uses a sample size of $n = 20\,000$. We repeat the same procedure but using the sample size $n = 1000$ in Fig. 7 and $n = 100\,000$ in Fig. 8. These three figures suggest that increasing the sample size indeed leads to a reduction of the absolute estimation error for both methods, which gradually approaches zero.

The left panel of Fig. 9 shows a pairwise comparison between the causal and reduced parametrisations and is based on the same samples as Fig. 3. Most of the points are located close to the identity line,

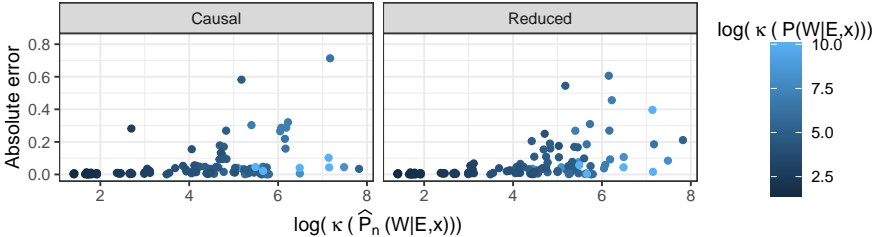

Figure 8: **Comparison of the absolute error for both parametrisations when having a large sample size.** Absolute error for a setting with $k_E = 2$, $n = 100\,000$, $M = 25$ and $N = 5$. The are more points close to zero than in Figs. 3 and 7, although there are still some estimates with a large absolute error.

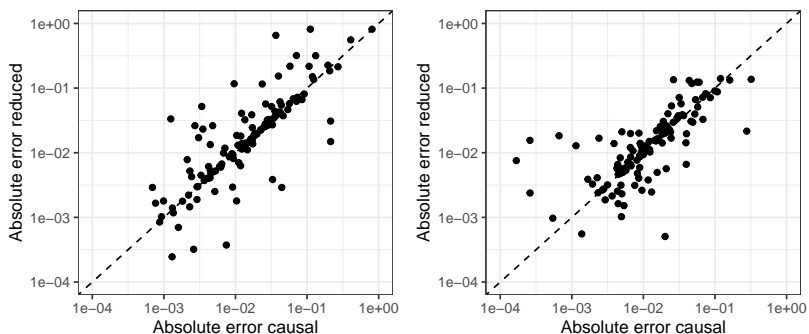

Figure 9: **Pairwise comparison of the absolute error for the causal and reduced parametrisations.** We compare the graphs with $k_E = 2$ (left, $n = 20\,000$) and $k_E = 3$ (right, $n = 25\,000$). The dashed line has a slope equal to 1. The results provided by both estimators are better for the causal estimator in the first case, while they become more similar when including one extra domain, where the reduced estimator provides slightly more accurate estimates, with an average absolute estimation error of 0.0255 for the causal estimator and 0.0252 for the reduced estimator.

so the absolute error is similar for both methods. The right panel shows the results when adding another source domain and shows smaller differences between the estimates of both parametrisations. Furthermore, we increase the sample size when adding an extra source domain to keep the average sample size per domain approximately constant in both settings.

Regarding the baseline comparison in Fig. 4, we only consider samples from induced distributions corresponding to causal models where $|q(y|\mathrm{do}(x)) - q(y|x)| > 0.1$ to avoid those cases in which the causal effect of $X$ on $Y$ was barely confounded. We compare the two estimators introduced in this work with the following five methods.

**Oracle.** We generate the data directly from the intervention distribution $\mathbb{Q}_Y^{\mathrm{do}(X:=x)}$. For an i.i.d. sample $Y_1, \ldots, Y_n$ from this marginal intervention distribution of $Y$ in the target domain, we estimate causal effect by

$$\widehat{q}_{\mathrm{oracle},n}(y|\mathrm{do}(x)) := \frac{1}{n} \sum_{i=1}^{n} \mathbb{1}(Y_i = y).$$

Since the oracle estimator has access to the intervention distribution, it should not be considered a competing method but rather a benchmark (or, as in § 6, an estimator for the ground truth). In the simulation experiments, the differences between the oracle estimates and the true value $q(y|\mathrm{do}(x))$ are due to the sample size being finite.

**No adjustment estimator.** This estimator ignores the hidden confounding and estimates the causal effect by estimating the conditional distribution $Y|X$. We consider two variants: the first one (NoAdj) uses an i.i.d. sample $(X_1, Y_1), \ldots, (X_n, Y_n)$ from the marginal distribution $\mathbb{P}_{(X,Y)}^{\mathcal{C}}$ pooled across all

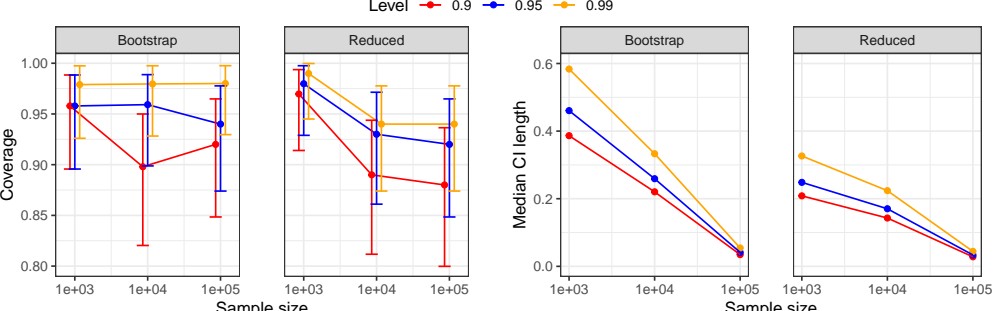

Figure 10: **Comparison of the coverage and median interval length of asymptotic and bootstrap confidence intervals as a function of the sample size.** The coverage is close to the nominal level in both cases but it is slightly better for the bootstrap method. The length of the bootstrap confidence intervals is larger than the ones obtained from the asymptotic normality of the estimator, especially for small values of the sample size.

source domains, while the second one (NoAdj*) uses an i.i.d. sample $(X_1, Y_1), \ldots, (X_n, Y_n)$ from the marginal distribution $\mathbb{Q}^{\mathcal{C}}_{(X,Y)}$ in the target domain; this, again, is information that is not available in our setting, so (NoAdj*) should not be considered a competing method, either. Both NoAdj and NoAdj* estimate the causal effect with the following expression:

$$\widehat{q}_{\text{NoAdj},n}(y|\text{do}(x)) := \frac{\sum_{i=1}^n \mathbb{1}(Y_i = y, X_i = x)}{\sum_{i=1}^n \mathbb{1}(X_i = x)}. \tag{27}$$

$W$**-adjustment estimator.** This estimator uses the adjustment formula (3) with the proxy variable $W$ instead of $U$. $\{W\}$ is not a valid adjustment set and, in general, this estimator is not a consistent estimator for the causal effect of $X$ on $Y$. However, if the proxy $W$ is 'similar' to the confounder $U$ (for example, if they are equal except for a small noise perturbation), the estimate could be close to the true value. We again consider two variants: one which uses an i.i.d. sample $(W_1, X_1, Y_1), \ldots, (W_n, X_n, Y_n)$ from $\mathbb{P}^{\mathcal{C}}_{(W,X,Y)}$ pooled across the source domains (WAdj) and the other from $\mathbb{Q}^{\mathcal{C}}_{(W,X,Y)}$, that is, from the target domain (WAdj*); as before, the latter estimator should not be considered a competing method as it uses information that is not available in our setting. In both cases, the estimator is defined as

$$\widehat{q}_{W\text{Adj},n}(y|\text{do}(x)) := \sum_{j=1}^{k_W} \left[ \left( \frac{\sum_{i=1}^n \mathbb{1}(Y_i = y, X_i = x, W_i = w_j)}{\sum_{i=1}^n \mathbb{1}(X_i = x, W_i = w_j)} \right) \left( \frac{\sum_{i=1}^n \mathbb{1}(W_i = w_j)}{n} \right) \right].$$

### C.4   Coverage of confidence intervals

We compare the proposed confidence intervals with bootstrap confidence intervals based on the normal approximation method [49]. Here, the variance is estimated using the sample variance $\widehat{\sigma}^2_B$ of the bootstrap estimates for $q(y|\text{do}(x))$. In that case, the bootstrap confidence interval is given by

$$\left[ \widehat{q}_{R,n}(y|\text{do}(x)) - \widehat{\sigma}_B \, z_{1-\frac{\alpha}{2}}, \widehat{q}_{R,n}(y|\text{do}(x)) + \widehat{\sigma}_B \, z_{1-\frac{\alpha}{2}} \right].$$

Fig. 10 presents the results comparing the bootstrap confidence intervals and the ones obtained through the asymptotic normality of the reduced estimator (see Eq. (13)). The comparison is done based on two properties of these intervals: coverage and median length. With regards to coverage, the empirical values obtained in the simulations are close to the nominal value, being slightly more accurate for the bootstrap method. However, the length of the bootstrap confidence intervals is larger.

### C.5   Runtime analysis

We compare the different estimation methods and baselines presented in Fig. 4 in terms of computation time. For this purpose, we measure the total time needed to compute an estimate 50 times (to obtain

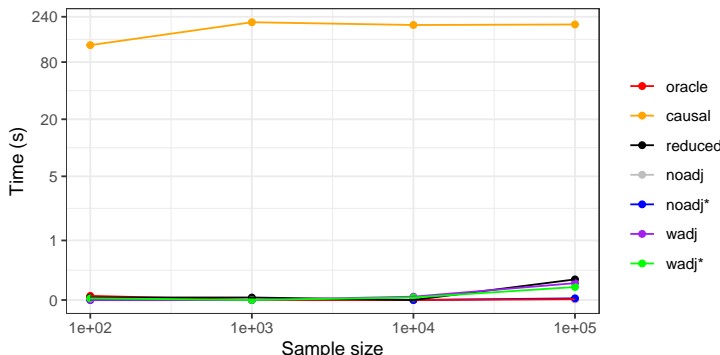

Figure 11: **Comparison of the computation time for the proposed methods and baselines.** The proposed estimators present a higher computation time than the baselines. However, the reduced estimator is only marginally slower than the $W$-adjustment estimators but presents a better performance in terms of absolute error in Fig. 4.

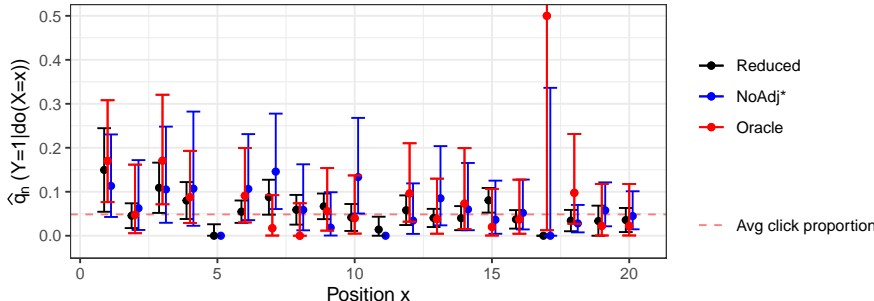

Figure 12: **Estimation of the causal effect of different ranking positions on the probability of clicking on a fixed hotel considered as target domain.** The horizontal dashed line represents the average proportion of clicks on that hotel across all positions. The confidence intervals for the positions 1 and 3 for the reduced and oracle estimators are completely above that line, showing that these positions have a significant effect on the probability of being clicked.

more accurate time measurements) with each estimator. This value depends on the implementation of each method and the equipment used for the measurement, so we focus on the comparison between the methods and not on the specific time values. Fig. 11 presents the computation time for each method as a function of the sample size. The proposed estimators have a computation time larger than the baselines, but this difference is marginal for the reduced estimator, staying below one second for this runtime analysis. The time is larger for the causal estimator due to the optimization procedure needed to calculate the estimate.

## D  Comparison of the effect of different ranking positions on user's choices

We estimate $q(Y = 1|\mathrm{do}(X = x))$ for different values of the position variable $x \in \{1, \ldots, 20\}$ in a unique fixed target domain, that is, the probability of clicking in a fixed hotel (we consider the hotel with target domain ID equal to 1 in the experiment in Fig. 5) under the atomic intervention of fixing its position to $x$. We again compare the reduced estimator with the oracle estimator. Fig. 12 shows the results. As before, there is an overlap between the confidence intervals for the reduced estimator and the oracle in all cases, except for those positions $x$ in which the position $X = x$ was not observed in the randomized dataset (positions 5 and 11), so it is not possible to calculate the oracle estimate. All the confidence intervals are at level 0.95 and the ones corresponding to the oracle and no adjustment estimators are calculated from the exact test for binomial data using the default `binom.test` function in R.

The horizontal line indicates the average proportion of clicks on that hotel across all positions when they are randomized; this allows us to see for which positions $x$ the probability of clicking on the hotel under the intervention $\mathrm{do}(X := x)$ is larger or smaller than the average value. The confidence intervals constructed using the reduced parametrisation and the oracle corresponding to the positions $X = 1$ and $X = 3$ are completely above the line representing the average proportion of clicks on the hotel across all positions. Therefore, we conclude that the probability of clicking on that specific hotel when its position is fixed to 1 or 3 through an intervention is significantly higher (at level 0.95) than the average proportion across all the positions.

