# OpenReview forum: "Transferring Causal Effects using Proxies"
_NeurIPS.cc/2025/Conference — NeurIPS 2025 poster_

### Official Review · Reviewer_qoW1 · 2025-06-02

**Clarity:** 3
**Significance:** 2
**Originality:** 2
**Rating:** 4
**Confidence:** 4

**Summary:**

This paper addresses the problem of estimating causal effects in a multi-domain setting. The authors leverage proxy variables to recover causal effects and propose an estimation method along with its theoretical properties. Finally, they present empirical evaluations and an application to demonstrate the practical utility of their approach.

**Questions:**

My concern is that the applicability of your result in practical settings may be substantially limited.

1. Your identification result states that the causal effect can be recovered from (W, X, Y) in the source domain and W in the target domain.
Are there any typos? I believe it should be (E, W, X, Y).

2. Your identification result appears to contradict my intuition. The causal effect depends on U, W, and N_Y, where both U and W follow different distributions in the source and target domains. It is therefore not intuitive that the causal effect could be recovered using only W.

3. The source of my concern lies in your specification of E and U. Your setting assumes that all factors responsible for shifts in the distribution of U across domains are fully observed through E. This is a very strong assumption, and if it is indeed being made, it should be stated explicitly and clearly in the text.
If N_U differs between the source and target domains, then your identification result does not hold. Is that correct? If that is the case, then the applicability of your result in practical settings may be substantially limited.

4. In relation to the concerns mentioned above, I also have reservations regarding the applicability of your results in practical settings.
In your application, E represents the hotel ID. Can all factors responsible for shifts in the distribution of U truly be captured by E alone? I do not think so.

Consider the following setting:

E: Hotel ID

N_{U1}: Customer’s age

N_{U2}: Customer’s gender

N_{U3}: Customer’s occupation

N_{U4}: Hotel location

N_{U5}: Quality of hotel service

The variables N_{U1} through N_{U5} may vary across different hotels. Consequently, the validity of your results appears questionable under this setting.
To ensure the validity of your results, it is necessary to observe all factors that account for shifts in the distribution of U, such as N_{U1} through N_{U5}.

5. Is it necessary to specify e_T for estimation? In the application, E represents hotel IDs, and the results change when the target hotel ID e_T is replaced. This behavior is quite strange.

**Ethical Concerns:**

["NO or VERY MINOR ethics concerns only"]

**Final Justification:**

My main concern has been resolved.

**Limitations:**

yes

**Quality:**

3

**Strengths And Weaknesses:**

**Strength**

This paper is mathematically rigorous.

**Waekness**

My concern is that the applicability of your result in practical settings may be substantially limited. (I will raise score if my concern is wrong.)

---

> ### Author Rebuttal · Authors · 2025-07-31
>
> ## General response to all reviewers and the AC
> We are grateful to all reviewers for their time and the useful feedback and suggestions. We are pleased that reviewers consider our work to tackle ”an important problem” (`hDXV`), “well-written” (`hDXV`, `nez5`), “mathematically rigorous” (`qoW1`), and found the empirical results “strong” (`nez5`) and “effective” (`aTiY`). Two reviewers also highlighted the proposal of new, provably consistent estimators as a key strength (`hDXV`, `aTiY`).
>
> A concern of reviewers `hDXV` and `qoW1` is the applicability of the proposed methods. For the former, we refer to Appx. B, where we provide relaxations for some of the conditions over the number of categories of $W$ and $U$. For the latter, we clarify in our response that our model can accommodate arbitrary shifts in the distribution of $U$, even without changes in $N_U$, via the domain indicator $E$. We thank reviewer `nez5` for pointers to related work on data fusion and transportability, which we will incorporate in Section 1.1. We will also include an evaluation of MAE for the WAdj and WAdj* baselines (for which no CIs are available), as well as a runtime analysis for the proposed methods to the baselines, following suggestions by reviewers `nez5` and `aTiY`, respectively.
>
> Our responses to individual reviewers provide further details. We believe to have thus addressed all reviewers’ concerns, but remain available for further discussion and clarification.
>
> ## Individual response to reviewer
> Thank you for your time in reviewing our work and for your questions and remarks which have helped us to clarify some key aspects of our setting.
>
> > Your identification result states that the causal effect can be recovered from (W, X, Y) in the source domain and W in the target domain. Are there any typos? I believe it should be (E, W, X, Y).
>
> Thank you for pointing out this possibly confusing formulation. What we mean here is that ``the causal effect can be recovered from $\{P(W,X,Y|E=e_1), …, P(W,X,Y|E=e_{k_E})\}$ and $Q(W)=P(W|do(E:=e_T))$’’, i.e., we do indeed require the domain labels. (The main point we wanted to convey here is that observations of $X$ and $Y$ are not needed in the target domain.) We will adjust the formulation accordingly to clarify.
> > Your identification result appears to contradict my intuition. The causal effect depends on U, W, and N_Y, where both U and W follow different distributions in the source and target domains. It is therefore not intuitive that the causal effect could be recovered using only W.
>
> It may perhaps be unintuitive at first, but this is precisely the main insight of proximal causal inference: a causal effect can be identified, even in the presence of unobserved confounding $U$, if sufficiently informative proxies $W$ are observed. This insight is not specific to our work, but also central to the works described in Section 1.1, many of which are by now well-established in the causal inference literature.
> While it is true that the joint and marginal distributions of $U$ and $W$ may change across domains, a crucial aspect of our setting is that the way in which $W$ depends on $U$---captured by the conditional distribution $P(W|U)$ (or $P(W|U,Z)$ for the more general setting discussed in Appx. B)---remains invariant across domains. This invariance (which follows from the modularity of causal models) is leveraged to transform the adjustment formula from Eq. (3) such that it only involves observable quantities, see also l.130-132.
> Imagine that $W$ is a perfect copy of $U$, i.e., $W=U$ and $P(W|U)$ is the identity. In this case, observing $W$ in the target domain suffices, because the terms in Eq. (3) become $P(y|W,x)$ and $Q(W)$. Our identification result shows that this argument extends to noisy or imperfect copies of $U$, provided that they are sufficiently informative as formalised by Asm. 1. In particular, Asm. 1 rules out cases such as that of a completely uninformative proxy $W$ that is independent of $U$, for which identification would become impossible.
> Intuitively, due to the invariance of other conditionals, observations of an informative proxy $W$ in the target domain can suffice to correct for changes in the distribution of $U$.
>
> > The source of my concern lies in your specification of E and U. Your setting assumes that all factors responsible for shifts in the distribution of U across domains are fully observed through E. This is a very strong assumption, and if it is indeed being made, it should be stated explicitly and clearly in the text. If N_U differs between the source and target domains, then your identification result does not hold. Is that correct? If that is the case, then the applicability of your result in practical settings may be substantially limited.
>
> We believe there may be a misunderstanding regarding our use of structural equations for modelling how and which distributions change across domains. We will attempt to clarify this key aspect of our setup.
>
> **SCMs and induced distributions.** The noises $N_i$ in Eq. (1) are exogenous (not being modelled) and only serve to introduce stochasticity into the otherwise deterministic model. The variables of interest are $E,U,W,X,Y$. Crucially, the functions $f_i$ and noise distributions $P_{N_i}$ are completely unconstrained. Hence, the SCM can induce any $P(E)$, $P(U|E)$, $P(W|U)$, $P(X|U)$, and $P(Y|U,W,X)$, see, e.g., Sec. 3.4 of Peters et al. [42] for details. In particular, the distributions of $U$ can change in arbitrary ways across domains, i.e., $P(U|E=e_1)$, …, $P(U|E=e_{k_E})$, $Q(U)$ are (a priori) unconstrained. While we use SCMs to formalize the data-generating process, the paper could also be written purely in terms of conditional pmfs (see l. 100-101).
>
> **The role of the environment variable $E$.** The variable $E$ is *not* assumed to explicitly capture “factors responsible for shifts in the distribution of U”. In line with prior literature on causal transportability and domain adaptation, $E$ is simply a **domain indicator** (e.g., $E \in \{1, 2, 3, 4\}$ for experiments run in different labs) that can be thought to switch between different distributions for $U$. We only assume that these distributions are stable within any given domain (i.e., for a fixed value of $E$), which is a standard assumption for many well-established methods. As explained above, the distributions of $U$ can, in general, differ arbitrarily across domains, even without $N_U$ changing; they are only implicitly constrained through Asm. 1.
>
> **Identifiability with Shifting Confounder Distributions.** To answer your direct question: ``If $N_U$ differs between the source and target domains, then your identification result does not hold. Is that correct?'' **No, this is incorrect.** Our identification result is agnostic to why the distribution of $U$ changes. The key requirement is that the rank condition, $\text{rank}(P(W|E,x)) \geq k_U$, is met in the source domains. The mechanism that generates $U$ can change arbitrarily. As explained above, the standard assumption that $N_U$ does not change across domains **comes without loss of generality**, but changes to $N_U$ could equally be accommodated without causing issues.
> We will revise the manuscript to clarify these crucial points and make the role of $E$ as a simple indicator variable and the relation between Eq. (1) and changes to $P(U)$ more explicit.
> In light of these explanations, we hope that you will agree that our modelling of the relationship between $E$ and $U$ *does not* limit the applicability of our results.
>
> > In relation to the concerns mentioned above, I also have reservations regarding the applicability of your results in practical settings. In your application, E represents the hotel ID. Can all factors responsible for shifts in the distribution of U truly be captured by E alone?
>
> We hope that our previous response also clarifies this issue.
> [To briefly reiterate the main point: Even without changes to the distribution of $N_U$, arbitrary changes in $P(U)$ can be modelled via $U:=f_U(E,N_U)$ via a domain indicator $E$. One way to see this is inverse cumulative distribution function (CDF) sampling. Consider two domains with arbitrary distributions $P_1(U)$ and $P_2(U)$. Let $N_U$ be uniform and define $f_U(e_1, n_U):=F_1^{-1}(n_U)$ and $f_U(e_2, n_U):=F_2^{-1}(n_U)$ where $F_1$ and $F_2$ are the CDFs of $P_1$ and $P_2$. This construction induces $P_1$ in $e_1$ and $P_2$ in $e_2$. The key is the flexibility of the function $f_U$, so that different values of $E$ can index different functions of the second (stochastic) argument.]
>
> > Is it necessary to specify e_T for estimation? In the application, E represents hotel IDs, and the results change when the target hotel ID e_T is replaced. This behavior is quite strange.
>
> $E$ is simply a categorical identifier for distinct domains. In the application, each hotel ID is an arbitrary label. The model does not use the value of the ID itself in any calculation.
> The key point is that changing the target ID from, say, Hotel A to Hotel B, means we are asking a different question: "What is the causal effect for Hotel A?" versus "What is the causal effect for Hotel B?". These environments have different characteristics—specifically, different distributions of the proxy variable $W$ (price). That is, changing the target hotel ID also changes $Q(W)$. Our method leverages this information from the proxy in the target domain to transport the causal effect correctly. Therefore, the estimate *should* change. If the results did not change when switching the target from a luxury hotel to a budget hotel, it would imply the model is failing to account for the target-specific context. The observed behavior thus confirms that the model is working as intended.
>
> **We hope that our responses have clarified some misunderstandings and resolved your concerns. In light of this, we kindly ask that you reconsider your evaluation.**

---

> > ### Comment · Reviewer_qoW1 · 2025-07-31
> > **Additional Questions**
> >
> > Sorry, my previous questions may have been slightly confusing. I also have some follow-up questions, phrased differently for clarity.
> >
> > 1. Graphically, in Figure 1(a), edges can appear from the environment variable E to other observed variables such as X, Y, and W. This is because changes in the environment are expected to influence not only the unobserved variables (e.g., U) but also the observed variables X, Y, and W directly. It is unclear whether the structure in Figure 1(a) reflects a realistic setting or serves as a simplified yet meaningful model. To me, it appears to have been introduced primarily for theoretical purposes. What justifies removing the edges from E to X, Y, and W?
> >
> > 2. If E is simply a domain indicator, then your identification results should be invariant to permutations of its labels. Can you formally establish or provide proof of this invariance property?
> >
> > 3. I am confused by the statement “The model does not use the value of the ID itself in any calculation,” because E appears explicitly in Theorem 1.

---

> > > ### Author Response · Authors · 2025-08-01
> > > **Reply to Additional Questions**
> > >
> > > Thank you for the continued engagement. We will happily address your follow-up questions.
> > >
> > > (1.) In general, it is correct that any aspects of the generative process could, in principle, change across domains, and that this would graphically manifest in edges from $E$ to any other variable. However, meaningfully learning from multiple domains requires that certain aspects are shared. Otherwise, if domains are completely unrelated, there is nothing to be gained from considering them jointly, and each domain should be analysed in isolation. To make progress and successfully leverage multi-domain data, it is therefore a standard assumption that part of the generative process remains invariant across domains. In causal approaches, this typically takes the form of a subset of invariant mechanisms or (conditional) distributions. Since the outcome $Y$ is not observed in the target domain of interest, it is clear that some (strong) assumptions are needed. Perhaps the most well-known such assumption is covariate shift, i.e., that only $P(X)$ changes and $P(Y|X)$ is the same. In our work, we instead exploit (a variation of) the latent shift assumption.
> > >
> > > Regarding our concrete setting, Fig. 1(a) only depicts **one example** of an SCM that satisfies our identifiability assumptions. However, our framework is *not* restricted to this specific graph structure. In particular, as detailed in Appx. C (especially Fig. 5), our assumptions apply to a **larger set** of SCMs, possibly containing some additional edges. For example,  $E$ can also affect $X$ directly, $W$ indirectly or additional observed covariates $Z$ directly. However, an edge $E \to Y$ is not allowed in our setting, as this would mean that the outcome model can change in unpredictable ways that cannot be correctly solely based on an observed proxy (without other additional assumptions).
> > >
> > > While our assumptions certainly may not hold for *all* real-world scenarios of interest, they are comparable in strength to those used by other work in the domain adaptation literature (e.g., [28], [30]).
> > >
> > > As for any causal inference method, whether the assumptions hold for any given specific setting or application needs to be critically discussed and decided on a case-by-case basis with the help of domain expertise—no method (including ours) is a silver bullet that applies in all scenarios. Nonetheless, as also highlighted by some of the other reviewers, we consider our work to address an important and useful subset of problems.
> > >
> > > (2.) Thank you for the suggestion. Indeed, the identification result is invariant to permutations of the (arbitrary) environment labels. This can be shown as follows.
> > >
> > > Consider the identification formula of Thm. 1 (explicitly indicating the expression for the pseudoinverse),
> > >
> > > $q(y|do(x)) = P(y|E,x)P(W|E,x)^\top(P(W|E,x)P(W|E,x)^\top)^{-1}Q(W)$.
> > >
> > > Each element $j$ of the product $P(y|E,x)P(W|E,x)^\top$ is $\sum_e p(y|e,x)P(w_j|e,x)$, while each element $(i,j)$ of $P(W|E,x)P(W|E,x)^\top$ is $\sum_e P(w_i|e,x)P(w_j|e,x)$. Therefore, values of $E$ only appear as indices of sums in the expression for $q(y|do(x))$. Since sums are invariant to permutation (i.e., $\sum_e = \sum_{\pi(e)}$ for any permutation $\pi$ of the indices), it follows that the identification result is invariant to permutations of the environment labels.
> > >
> > > We will add this fact to our (updated) explanation of the domain indicator variable in Sec. 2.
> > >
> > > (3.) Apologies for the confusion. What we mean is that “the precise encoding of domains does not matter”: because $E$ is a categorical random variable, how we label its categories is arbitrary. For example, a setting with three hotels is equally well represented by {$1,2,3$}, {$HotelA, HotelB, HotelC$}, or any other set of three *distinct* strings. That is, the category labels are not meaningful (provided they can be distinguished). Only the (conditional) probabilities defined in terms of these categories matter.
> > >
> > > In particular, in Thm. 1, $E$ only appears as a conditioning variable inside conditional probabilities. It serves to define the domain-specific distributions over the other variables, which are then used in the identification formula (e.g., [30] uses a subscript notation instead of a conditional notation, so $p_1(Y=y|X=x) = p(Y=y|X=x,E=e_1)$, referring to the conditional distribution of $Y$ given $X=x$ in a domain arbitrarily labelled as $e_1$).
> > > In particular, the ID of the target hotel is only used to find the correct distribution of the proxy $Q(W)$ to plug into the identification formula.
> > >
> > > Please let us know if this answers your questions or whether some points remain unclear.

---

> > > > ### Comment · Reviewer_qoW1 · 2025-08-03
> > > >
> > > > Thank you for your clarification. I will raise the score to 4, but I think more clarification of the setting is needed.
> > > >
> > > > 1. Based on your settings, which mechanisms are variable and which are invariant to the environment? Which conditional causal effects are invariant to the environment? As with the distributional literature, lists of variant and invariant sets of probabilities are helpful. For example, Covariate shift: The distribution of input variables P(X) changes, but the conditional distribution P(Y|X) remains the same.
> > > >
> > > > 2. How can you justify the setting in your application?

---

> ### Author Response · Authors · 2025-08-04
> **Additional clarification of the setting**
>
> Thank you for the continued feedback and for reconsidering your evaluation. We address your remaining questions below.
>
> > Based on your settings, which mechanisms are variable and which are invariant to the environment? Which conditional causal effects are invariant to the environment? As with the distributional literature, lists of variant and invariant sets of probabilities are helpful. For example, Covariate shift: The distribution of input variables P(X) changes, but the conditional distribution P(Y|X) remains the same.
>
> The mechanisms that may vary are those for which $E$ appears as an argument on the RHS of the corresponding structural equation in Eq. (1); the others remain invariant. (This follows from the Markov property, see l.99-100.) In terms of the induced conditional probability distributions, this means that:
> - $P(U)$, or rather $P(U|E)$, varies across domains
> - $P(X|U)$, $P(W|U)$, and $P(Y|W,U,X)$ remain invariant
>
> While this may seem like a rather strong assumption at first, it is important to consider that $U$ is *unobserved*. When restricting our attention to observable variables (as is common practice), there are *no invariances*, i.e., the entire joint distribution $P(X,Y,W)$ can vary across domains. In particular, $P(X)$, $P(Y|X)$, $P(Y)$, and $P(X|Y)$ may all change. In this sense, the latent shift assumption is weaker than, e.g., covariate shift, label shift, or conditional shift (which all require some form of invariance involving only observed variables).
>
> Regarding invariant conditional causal effects: Since only $P(U)$ changes across domains, any causal effect that conditions on $U$ will be domain-invariant. However, because $U$ is unobserved, this does not have direct practical implications. If we again consider conditional causal effects involving only *observed* variables, generally none of them are domain-invariant. This precisely necessitates our approach for correcting or transporting conditional causal effects to the target domain.
>
> We will try to make these points clearer in the updated descriptions of prior work (Sec. 1.1) and our setting (Sec. 2).
>
>
> > How can you justify the setting in your application?
>
> As mentioned in your initial review, many factors could conceivably confound the causal effect of interest, such as the location or service quality of the hotel. Our setting allows for shifts in the distribution of these factors (as summarised in the form of a categorical variable $U$) but requires that the other conditional distributions given $U$ remain invariant. For example, invariance of $W|U$  implies that different hotels that share the same characteristics $U=u$ (location, service quality, etc) will also exhibit the same price distribution.
>
> Our assumptions from Fig. 1(a) likely do not perfectly match the setting at hand. As shown in Appx. C.2, though, our results also hold for more general graphs with additional covariates or edges, which may be more appropriate. Our objective was to evaluate the developed estimators for the simple setting presented in the main paper. Our empirical results from Fig. 4 and the MAEs reported in our response to reviewer `nez5`confirm the effectiveness of our estimator compared to baselines, despite likely misspecification and violations of some of our assumptions.

---

### Official Review · Reviewer_nez5 · 2025-06-25

**Clarity:** 4
**Significance:** 3
**Originality:** 3
**Rating:** 5
**Confidence:** 3

**Summary:**

The paper studies a setting where data has been collected from multiple domains and we are interested in estimating the interventional distribution in a specific target domain. In this setting, unmeasured confounding is allowed but to our help we have an outcome-related proxy. The authors propose two novel estimators which require information about the treatment, outcome and proxy in the source domains and only the proxy in the target domain. They prove the estimators are consistent, and asymptotic normality for one of them. In a set of experiments, they show how the proposed estimators are able to reduce the bias from the unmeasured confounding relative to a set of baseline estimators which ignores the confounding or adjusts for the proxy as if it was the confounder.

**Questions:**

1. Why is not WAdj and WAdj* not included in the real data experiment?
2. In the real data experiment, for some cases the reduced estimators disagrees with the oracle and the NoAdj is actually closer to the oracle, for instance target domain id 3 or 11. Do you have an hypothesis for why this happens in this particular cases? I wonder if any of the assumptions you make are particularly violated here.

**Ethical Concerns:**

["NO or VERY MINOR ethics concerns only"]

**Final Justification:**

The authors satisfactorily addressed my concerns. I stand by the same score as before, which was 5. Happy to recommend this paper for an accept to the conference.

**Limitations:**

Yes.

**Paper Formatting Concerns:**

None found.

**Quality:**

3

**Strengths And Weaknesses:**

# Strengths
- The paper is very well-written overall and has an easy structure to follow.
- I liked the prior works section in the beginning and it helped me understand the context of this paper.
- The empirical results seem strong, in particular the real data experiment.

# Weaknesses
- It is not clear to me why WAdj and WAdj* are missing from the real data experiment, I would apriori have thought that it would make more sense to compare to than the NoAdj and NoAjd* since it showed a lower absolute error in the other simulations.
- While the paper generally does a nice job mentioning relevant related works, I feel like it is overlooking important work on e.g. data fusion in causal inference. I understand that the domain adaption papers are closely related, but there also exists a rich literature in causality on the multi-domain setting as well. It would be great if the authors also could relate their setting and method to these papers; I've added some references as examples.

#### References
Bareinboim, Elias, and Judea Pearl. "Causal inference and the data-fusion problem." Proceedings of the National Academy of Sciences 113.27 (2016): 7345-7352.
Dahabreh, Issa J., and Miguel A. Hernán. "Extending inferences from a randomized trial to a target population." European journal of epidemiology 34 (2019): 719-722.
Colnet, Bénédicte, et al. "Causal inference methods for combining randomized trials and observational studies: a review." Statistical science 39.1 (2024): 165-191.

---

> ### Author Rebuttal · Authors · 2025-07-31
>
> ## General response to all reviewers and the AC
> We are grateful to all reviewers for their time and the useful feedback and suggestions. We are pleased that reviewers consider our work to tackle ”an important problem” (`hDXV`), “well-written” (`hDXV`, `nez5`), “mathematically rigorous” (`qoW1`), and found the empirical results “strong” (`nez5`) and “effective” (`aTiY`). Two reviewers also highlighted the proposal of new, provably consistent estimators as a key strength (`hDXV`, `aTiY`).
>
> A concern of reviewers `hDXV` and `qoW1` is the applicability of the proposed methods. For the former, we refer to Appx. B, where we provide relaxations for some of the conditions over the number of categories of $W$ and $U$. For the latter, we clarify in our response that our model can accommodate arbitrary shifts in the distribution of $U$, even without changes in $N_U$, via the domain indicator $E$. We thank reviewer `nez5` for pointers to related work on data fusion and transportability, which we will incorporate in Section 1.1. We will also include an evaluation of MAE for the WAdj and WAdj* baselines (for which no CIs are available), as well as a runtime analysis for the proposed methods to the baselines, following suggestions by reviewers `nez5` and `aTiY`, respectively.
>
> Our responses to individual reviewers provide further details. We believe to have thus addressed all reviewers’ concerns, but remain available for further discussion and clarification.
>
> ## Individual response to reviewer
> Thank you for your time reviewing our work and your suggestions to extend our section about related literature. We address each point individually below.
>
> > It is not clear to me why WAdj and WAdj* are missing from the real data experiment.
>
> The reason for excluding WAdj and WAdj* from the real data experiment was the lack of a method for calculating confidence intervals for these estimators, which would reduce the relevance of the comparison presented in Figures 4 and 10. Wald or exact binomial test confidence intervals cannot be used for WAdj and WAdj*, as opposed to the Oracle and NoAdj* estimators, and there is not a simple derivation as in the case of the reduced estimator. The main alternative would be to consider bootstrap confidence intervals.
> To compare the point estimates of WAdj and Wadj* with the rest of the methods in the real data experiment, we have computed the mean absolute error (MAE) with respect to the oracle during this rebuttal period obtaining the following values: 0.044 (Reduced), 0.051 (NoAdj), 0.080 (NoAdj*), 0.053 (WAdj) and 0.075 (WAdj*). This suggests that the proposed reduced estimator performs better in terms of absolute error than all the considered baselines in this real data example. We will include these results in Section 6 of the main paper.
>
> > I feel like it is overlooking important work on e.g. data fusion in causal inference.
>
> Thank you very much for the pointers. We agree that prior works on data fusion and transportability, combining observational and experimental studies, and causal approaches to multi-domain data are related and will include additional paragraphs in Sec. 1.1. To briefly comment on some commonalities and differences:
> The work of Pearl, Bareinboim et al. on data fusion considers a similar problem. However, in that line of work, transportability is typically determined purely from the extended causal graph (a.k.a. selection diagram) and (extensions of) do-calculus. In contrast, proximal causal inference is less fully nonparametric and typically relies on additional constraints such as Asm. 1 (e.g., requiring invertibility of certain matrices, thus formalising sufficiently informative proxies). In our setting, identifiability thus cannot be concluded solely from the causal graph, but leverages additional assumptions of a different nature than those explored in the transportability literature.
> Work on combining experimental and observational data typically aims to overcome the statistical challenge of small sample size in the experimental study, e.g., by devising biased but lower variance estimators that incorporate observational data. In contrast, in our setting, we do not have any observations of $X$ or $Y$ in the target domain (taking the role of the experimental data), but only observational data from different domains.
> Causal methods for multi-domain data often seek to learn an invariant predictor by discarding spurious features. Like the most closely related work by Tsai et al., they typically target the conditional mean $E_Q[Y|x]$, rather than the interventional distribution $q(y|do(x))$.
>
> > In the real data experiment, for some cases the reduced estimators disagrees with the oracle and the NoAdj is actually closer to the oracle, for instance target domain id 3 or 11. Do you have an hypothesis for why this happens in this particular cases? I wonder if any of the assumptions you make are particularly violated here.
>
> The NoAdj and NoAdj* estimators do not consider the existence of an unmeasured confounder $U$. Therefore, in those domains in which the causal effect of $X$ on $Y$ is barely confounded, these estimators can obtain good estimates in terms of absolute error. Even though the change of domain does not directly affect the equations for $X$ and $Y$ in the SCM, the strength of the confounding can change due to the distribution shifts of $U$ when changing the domain. Furthermore, the NoAdj* estimator, which is the one that performs really well in terms of absolute error for the target domains ids 3 and 11 uses an i.i.d. sample $(X_1,Y_1),\dots,(X_n,Y_n)$ from the target domain, and this information is not available in our setting. Therefore, NoAdj* has access to some data that our proposed estimators do not (see Appendix D.3 for additional details).

---

> > ### Comment · Reviewer_nez5 · 2025-08-03
> >
> > I thank the authors for their response to my questions, which I believe they have addressed satisfactorily. I continue to stand by my current score of 5 and recommend acceptance of the paper.

---

### Official Review · Reviewer_aTiY · 2025-06-30

**Clarity:** 3
**Significance:** 3
**Originality:** 3
**Rating:** 5
**Confidence:** 3

**Summary:**

The paper proposes an approach on identifying and estimating the causal effect of $X$ on $Y$ in the target domain, by using the proxy variable $W$ of latent confounder $U$ affecting $X$ and $Y$ in multiple-domain settings. Also, the work demonstrates its application through experiments on real-world and synthetic datasets.

**Questions:**

* Suggestions

It could be useful to have a general runtime analysis, in a form of $O()$ analysis (if applicable), or across different controlled variables (e.g., $n$) and compare the proposed estimators against other baseline estimators, rather than on one specific instance as shown in Footnote 5, page 7.

* Questions

Is the identification formula shown in Eq. (4) of Theorem 1 complete? In other words, is it the case that $W$ is the proxy variable (as defined in Assumption 1) if and only if $q(y|do(x))$ is identified with Eq. (4)?

**Ethical Concerns:**

["NO or VERY MINOR ethics concerns only"]

**Final Justification:**

I am satisfied with the authors addressing my main concerns.

**Limitations:**

Yes

**Quality:**

3

**Strengths And Weaknesses:**

* Strengths
1. Identifying $q(y|do(x))$ with $W$ does not require a fully specified causal graph.
2. The work provides (economical) estimator of $q(y|do(x))$ and show that those are consistent (Propositions 2 ~ 4). Asymptotic confidence intervals for $q(y|do(x))$ is also given (Eq. (13)).
3. Experiments performed over real-world and synthetic datasets show effectiveness of the proposed estimators.

* Weaknesses
1. The paper does not seem to provide a method on finding any valid $W$ if one exists (in polynomial time), or a method on enumerating all valid $W$ (although it may be a different research problem).

---

> ### Author Rebuttal · Authors · 2025-07-31
>
> ## General response to all reviewers and the AC
> We are grateful to all reviewers for their time and the useful feedback and suggestions. We are pleased that reviewers consider our work to tackle ”an important problem” (`hDXV`), “well-written” (`hDXV`, `nez5`), “mathematically rigorous” (`qoW1`), and found the empirical results “strong” (`nez5`) and “effective” (`aTiY`). Two reviewers also highlighted the proposal of new, provably consistent estimators as a key strength (`hDXV`, `aTiY`).
>
> A concern of reviewers `hDXV` and `qoW1` is the applicability of the proposed methods. For the former, we refer to Appx. B, where we provide relaxations for some of the conditions over the number of categories of $W$ and $U$. For the latter, we clarify in our response that our model can accommodate arbitrary shifts in the distribution of $U$, even without changes in $N_U$, via the domain indicator $E$. We thank reviewer `nez5` for pointers to related work on data fusion and transportability, which we will incorporate in Section 1.1. We will also include an evaluation of MAE for the WAdj and WAdj* baselines (for which no CIs are available), as well as a runtime analysis for the proposed methods to the baselines, following suggestions by reviewers `nez5` and `aTiY`, respectively.
>
> Our responses to individual reviewers provide further details. We believe to have thus addressed all reviewers’ concerns, but remain available for further discussion and clarification.
>
> ## Individual response to reviewer
> Thank you very much for your time. We answer your individual questions below.
> > The paper does not seem to provide a method on finding any valid W if one exists (in polynomial time), or a method on enumerating all valid W.
>
> Thanks for the remark. We will address what we consider as two separate aspects: choosing among known proxies and finding new ones.
>
> Regarding how to select the best proxy from a known set of valid proxies, our framework does not require a selection. Instead, these proxies can be combined into a single, more informative proxy (i.e., consider the combined proxy that takes values in the Cartesian product of the sets of values the individual proxies can take). If any of the proxies satisfy Asm.  2, then the combined proxy will satisfy it as well. Therefore, the problem of selection is sidestepped by combination.
>
> Regarding how to efficiently find or enumerate all valid proxies $W$ in a *known* large, complex causal graph, we agree with the reviewer that this is an interesting and distinct research problem. The focus of our paper is on establishing identifiability and estimation given a valid proxy. Developing a polynomial-time search algorithm to check the graphical conditions listed in Remark 8 in Appx. B is a valuable direction for future work but falls outside the scope of our current contribution.
> If the causal graph is completely unknown, it does not appear to be testable whether a valid proxy exists. Instead, domain expertise is needed to argue for the validity of a given proxy (similar to justifying instrument validity in IV models).
>
> > It could be useful to have a general runtime analysis, in a form of O() analysis (if applicable), or across different controlled variables (e.g., n) and compare the proposed estimators against other baseline estimators, rather than on one specific instance as shown in Footnote 5, page 7.
>
> Thank you for the suggestion. Since our main focus was not on computational cost of the proposed methods, we only included a specific instance (Footnote 5) to show that it is not really computationally expensive. The main factor that increases the runtime is the dimension (number of categories) of the categorical random variables. However, we will look into this further and will try to augment the manuscript with a O() analysis or more comprehensive runtime numbers.
>
>
>
> > Is the identification formula shown in Eq. (4) of Theorem 1 complete? In other words, is it the case that W is the proxy variable (as defined in Assumption 1) if and only if q(y|do(x)) is identified with Eq. (4)?
>
> Yes, we believe the identification formula in Equation (4) to be complete. Thm. 1 shows that $q(y|do(x))$ is identified under Assumption 1. Conversely, if $q(y|do(x))$ is identified with Eq. (4) for any form of the confounder, then the right pseudoinverse of $P(W|E,x)$ exists and, therefore, this matrix has linearly independent rows. Furthermore, in this case, it can be shown that the inequality $k_W \geq k_U$ is obtained as follows from Equation (4):
>
> $\forall Q(U), P(y|U,x): \quad P(y|U,x) Q(U) = P(y|E,x) P(W|E,x)^{\dagger} P(W|U) Q(U) \Rightarrow$
>
> $\forall  P(y|U,x): \quad P(y|U,x) = P(y|E,x) P(W|E,x)^{\dagger} P(W|U) \Rightarrow $
>
> $\forall  P(y|U,x): \quad P(y|U,x) = P(y|U,x) P(U|E,x) P(W|E,x)^{\dagger} P(W|U) \Rightarrow $
>
> $I_{k_U}= P(U|E,x) P(W|E,x)^{\dagger} P(W|U) \Rightarrow $
>
> $k_U = rank(I_{k_U}) \leq \min( rank(P(U|E,x)), rank(P(W|E,x)), rank(P(W|U)) ) \leq k_W $

---

> > ### Comment · Reviewer_aTiY · 2025-08-06
> >
> > > Regarding how to efficiently find or enumerate all valid proxies $W$ [...]
> >
> > Thank you for sharing your perspective regarding these problems.
> >
> > > Since our main focus was not on computational cost of the proposed methods [...]
> >
> > Yes, I understand. But some readers may be interested in applicability aspect as well (other reviewers had pointed out different types of applicability).
> >
> > > Yes, we believe the identification formula in Equation (4) to be complete
> >
> > I think that completeness result would make the result stronger, and thus it may be good to emphasize it, e.g., 'sound and complete' identification.
> >
> > Overall, I am satisfied with my concerns being addressed in the reply.

---

### Official Review · Reviewer_hDXV · 2025-07-05

**Clarity:** 3
**Significance:** 2
**Originality:** 2
**Rating:** 4
**Confidence:** 3

**Summary:**

The authors consider the problem of causal effect identification of variable $X$ on $Y$ in the target domain under the presence of an unobserved confounder and proxy variable $W$ from multiple-domain observations (source domain). In their setting, the SCM is the same across the domains except for the equation for the unobserved variable $U$ that depends on the environment. The main estimand in this work is $P(y|do(x))$ in the target domain, that is  by knowing it one can identify Average Treatment Effect or Conditional Average Treatment Effect.

Under the specific assumption on the rank of matrix $P(W|E, x)$ and domains of variables $U$, $W$, $E$, authors propose an identification result for  $P(y|do(x))$. Further, they proposed two estimation procedures based on different decompositions of $P(y|do(x))$. Finally, the authors propose the theoretical results that the estimators are consistent.

**Questions:**

See Strengths And Weaknesses.

**Ethical Concerns:**

["NO or VERY MINOR ethics concerns only"]

**Final Justification:**

Most of my concerns raised by the review were positively addressed by the authors and were resolved to some extent. Therefore I am raising my score just to the 4.

**Quality:**

2

**Strengths And Weaknesses:**

Strength:
The authors consider an important problem of causal effect identification of $X$ on $Y$ under the presence of a latent variable $U$ and one proxy variable $W$. To enable identification results, authors assume access to the observational data from multiple environments. Access to the observational data from multiple environments substitutes the need for knowledge of the distribution of $P(W|U)$ as was done in Kuroki and Pearl [12]. Further authors propose two estimation procedures based on different decompositions of $P(y|do(x))$ and proved that the proposed estimation algorithms are consistent.

Also, the paper is well-written.

Weaknesses:
- To enable identifiability of $P(y|do(x))$ in the target domain, the authors require that the number of domains $k_E$ is bigger than the number of realizations $k_W, k_U$ of variables $W$ and $U$, respectively and additionally $k_W$ should be equal to $k_U$. This assumption is unrealistic, and almost never happens in practice. Therefore, I am not sure how significant the result is.
- How does the estimator that is presented in Section 4.1 relate to Theorem 1?
- Is the identifiability of $P(y|do(x))$ in Section 4.1 also requires the assumption 1?
- The assumption that only the equation of unobserved variables changes across the environments is also seemed to be too strong and rarely observed in practice.

---

> ### Author Rebuttal · Authors · 2025-07-31
>
> ## General response to all reviewers and the AC
> We are grateful to all reviewers for their time and the useful feedback and suggestions. We are pleased that reviewers consider our work to tackle ”an important problem” (`hDXV`), “well-written” (`hDXV`, `nez5`), “mathematically rigorous” (`qoW1`), and found the empirical results “strong” (`nez5`) and “effective” (`aTiY`). Two reviewers also highlighted the proposal of new, provably consistent estimators as a key strength (`hDXV`, `aTiY`).
>
> A concern of reviewers `hDXV` and `qoW1` is the applicability of the proposed methods. For the former, we refer to Appx. B, where we provide relaxations for some of the conditions over the number of categories of $W$ and $U$. For the latter, we clarify in our response that our model can accommodate arbitrary shifts in the distribution of $U$, even without changes in $N_U$, via the domain indicator $E$. We thank reviewer `nez5` for pointers to related work on data fusion and transportability, which we will incorporate in Section 1.1. We will also include an evaluation of MAE for the WAdj and WAdj* baselines (for which no CIs are available), as well as a runtime analysis for the proposed methods to the baselines, following suggestions by reviewers `nez5` and `aTiY`, respectively.
>
> Our responses to individual reviewers provide further details. We believe to have thus addressed all reviewers’ concerns, but remain available for further discussion and clarification.
>
> ## Individual response to reviewer
> Thank you for your time in reviewing our work. We respond to your questions and comments below.
>
> > To enable identifiability of P(y|do(x)) in the target domain, the authors require that the number of domains k_E is bigger than the number of realizations k_U, k_W of variables W and U, respectively and additionally k_W should be equal to k_U. This assumption is unrealistic, and almost never happens in practice. Therefore, I am not sure how significant the result is.
>
> Thank you for this question. It addresses an important part of our work. We agree that Assumption 1, which implies $k_W = k_U$, is restrictive. However, this is a simplified version of our assumption, presented in the main text for clarity. We provide a **more general and weaker assumption in Appx. B** (we mention this in Sec. 3.1). To avoid confusion, we decided that for the revised version of our manuscript, we will only state the general result of Appx. B directly in the main part of the paper.
>
> **Clarification of the Necessary Assumption.** Our results only require the rank condition $rank(P(W|E,x)) \geq k_U$ to hold (see Appx. B). This condition is less restrictive than the one highlighted by the reviewer and only implies that $k_E,  k_W \geq k_U$. Importantly, this rank condition is also a **necessary** condition for identifiability. If $rank(P(W|E,x)) < k_U$, it is, in general, not possible to identify the causal effect. To illustrate, consider the case where $E$ and $W$ are only correlated with the first $k_U - 1$ values the confounder $U$ can take.  In this scenario, the last value, $u_{k_U}$, can act as a hidden confounder with no proxy, making $P(Y | \text{do}(X))$ non-identifiable. More precisely, assume we have an SCM with hidden variable $\tilde{U}$ such that $rank(P(W|E,x)) = k_U-1$. Now modify this hidden confounder such that $U \coloneqq \tilde{U} 1(N=1) + u_{k_U} 1(N \neq 1)$, where $N$ is independent of $E$ and $W$, to get a new SCM. If in the new SCM $X$ and $Y$ are correlated through $U = u_{k_U}$, then this confounding cannot be detected by $E$ or $W$ as the joint distribution of $(E, W)$ does not change between the two SCMs. More generally, if the rank condition is violated, there will always be a linear combination of the confounder's values whose influence on $Y$ and $X$ cannot be detected through $E$ or $W$. We will formalize this intuitive argument in the revised manuscript.
>
> **Significance.** Achieving identifiability in the presence of unobserved confounders is a fundamentally challenging problem that always requires strong assumptions. We contend that our more general assumptions in Appendix B, particularly the generalized rank condition, is no stronger than those made in comparable works that achieve identifiability. In the revised version of our manuscript, we will expand our discussion to better contextualize our assumptions within the broader literature.
>
> > How does the estimator that is presented in Section 4.1 relate to Theorem 1?
>
> Thm. 1 is a theoretical result establishing that the causal effect is identifiable under our assumptions. While the theorem's statement directly suggests a "plug-in" estimator (i.e., by estimating the matrices on the right-hand side), such an approach is not necessarily the only possible estimator. We therefore also propose an estimator based on Maximum Likelihood of the full causal structure, building off Prop. 2. The connection to Thm. 1 is that our proof of consistency for this estimator (Prop. 3) requires the identifiability result from Thm. 1. In short, the theorem guarantees a unique target exists, which our proposed estimator is proven to converge to.
>
> > Is the identifiability of P(y|do(x)) in Section 4.1 also requires the assumption 1?
>
> As discussed in our answer to the previous question, the identifiability result presented in Sec. 3 is required for the consistency result of the estimators in Sec. 4.1. For the revised manuscript, we will make this connection clearer.
>
> > The assumption that only the equation of unobserved variables changes across the environments is also seemed to be too strong and rarely observed in practice.
>
> We thank the reviewer for this comment, as it highlights a point we need to make clearer in the paper.
> Our method is **not** limited to the setting where only the unobserved variables change. This was a simplifying assumption used for the illustrative example in the main text, chosen for pedagogical clarity and its connection to previous work [30].
> In fact, our theoretical guarantees hold for a much broader class of SCMs. In our manuscript we provide a detailed discussion and examples of these more general cases---where, for example, the mechanisms for observed variables like $X$ also shift across environments---in Appendix C.2 (Figure 5). The key requirement for our method is not the immutability of specific equations, but rather the structural properties that allow for identification.

---

> ### Comment · Reviewer_hDXV · 2025-08-04
>
> I thank the authors for their responses; most of my concerns were resolved, and I will increase my score to 4.
> Also, I will give a few comments that I have after reading the author's response:
> - Even if $k_W=k_U$ is just a simplified version of the assumption presented in the main work, still, my main concern was regarding the assumption $k_E \geq k_U$. Via simple words, it requires observing the data in the different domains/environments, the number of which is not less than the size of the domain of the unobserved variable $U$. This solely sounds to me very unrealistic. But it is nice to have a theoretical result that claims that this is a necessary condition for the identifiability.
> - `Our method is not limited to the setting where only the unobserved variables change.` This is very confusing, because the equations (1) at the beginning of Section 2 define the data generating process in which only the generating process of $U$ is domain dependent. If this is only a simplification, I would ask the authors to make it more explicit in the main part or at least make some reference to the Appendix, where this assumption is simplified.

---

> > ### Author Response · Authors · 2025-08-05
> >
> > Thank you for raising your score and for the additional feedback, which we address below.
> >
> > > Even if $k_W=k_U$ is just a simplified version of the assumption presented in the main work, still, my main concern was regarding the assumption $k_E \geq k_U$. Via simple words, it requires observing the data in the different domains/environments, the number of which is not less than the size of the domain of the unobserved variable $U$. This solely sounds to me very unrealistic. But it is nice to have a theoretical result that claims that this is a necessary condition for the identifiability.
> >
> > We agree that requiring $k_E \geq k_U$ can be a limiting factor for some applications, particularly when the source of confounding is thought to be complex and only few domains are available. As you rightfully point out, however, the necessity argument from our response to reviewer `aTiY` shows that identification via Thm. 1 is not possible in this case.
> >
> > At the same time, we believe that $k_E \geq k_U$ can still be a reasonable assumption for some practical scenarios.  For example, in the Expedia Hotel Searches dataset [39], the number of available domains (i.e., hotels) is actually very large ($>100,000$). In our case study from Sec. 6, we only use the 25 largest domains (hotels appearing in $>2000$ searches), but we could easily obtain hundreds or even thousands of domains by reducing the minimum sample size (number of searches in which a hotel appears). However, the agreement of our estimator with the oracle suggests that even $k_E=25$ may not be too unrealistic.
> >
> > > “Our method is not limited to the setting where only the unobserved variables change.” This is very confusing, because the equations (1) at the beginning of Section 2 define the data generating process in which only the generating process of $U$ is domain dependent. If this is only a simplification, I would ask the authors to make it more explicit in the main part or at least make some reference to the Appendix, where this assumption is simplified.
> >
> > We apologise for the confusion.
> >
> > It is correct that the setting presented in Sec. 2 and Eq. (1) is based on the latent shift assumption, which states that only the mechanism for $U$ is domain-dependent. For this basic setting, we not only show identifiability (Sec. 3) but also propose two consistent estimators with confidence intervals (Sec. 4) and conduct simulations and a case study (Secs. 5 & 6).
> >
> > However, our identifiability results from Sec. 3 can actually be generalized to a much broader setting in which the mechanisms for $X$ and additional covariates $Z$ also change across domains. This generalization is presented in detail in Appx. C, see Fig. 5 for some possible graphs, Remark 8 for the precise graphical conditions, and Thm. 7 for the generalized identification result.
> >
> > We agree that the current presentation, particularly regarding the highlighted comment, may be confusing. We will add pointers to the relevant appendices and clarify this aspect in the revised manuscript.

---

### Decision · Program_Chairs · 2025-09-17

**Decision:**

Accept (poster)

**Comment:**

This paper addresses the problem of estimating causal effects in a multi-domain setting with unobserved confounders, proposing a methodology to identify and estimate the target domain's causal effect using proxies of the hidden confounder, with theoretical guarantees of identifiability, consistent estimators, and supporting empirical results.

All reviewer's main concerns were thoroughly addressed: authors clarified assumptions to be more general than initially perceived, strengthened theoretical connections and supplemented experimental details. The responses effectively resolved doubts about applicability, theoretical rigor, and experimental validity. Considering the paper, reviews and discussion overall, I recommend this paper for acceptance.